# Transcriptome size matters for single-cell RNA-seq normalization and bulk deconvolution

Songjian Lu, Jiyuan Yang [ID], Lei Yan [ID], Jingjing Liu [ID], Judy Jiaru Wang, Rhea Jain & Jiyang Yu [ID] [✉]

The variation of transcriptome size across cell types significantly impacts single-cell RNA sequencing (scRNA-seq) data normalization and bulk RNA-seq cellular deconvolution, yet this intrinsic feature is often overlooked. Here we introduce ReDeconv, a computational algorithm that incorporates transcriptome size into scRNA-seq normalization and bulk deconvolution. ReDeconv introduces a scRNA-seq normalization approach, Count based on Linearized Transcriptome Size (CLTS), which corrects differential expressed genes typically misidentified by standard count per 10 K normalization, as confirmed by orthogonal validations. By maintaining transcriptome size variation, CLTS-normalized scRNA-seq enhances the accuracy of bulk deconvolution. Additionally, ReDeconv mitigates gene length effects and models expression variances, thereby improving deconvolution outcomes, particularly for rare cell types. Evaluated with both synthetic and real datasets, ReDeconv surpasses existing methods in precision. ReDeconv alters the practice and provides a new standard for scRNA-seq analyses and bulk deconvolution. The software packages and a user-friendly web portal are available.

The intricate arrangement of different cell types within tissues is essential for tissue functionality and homeostasis[1]. Changes in cellular composition often correlate with disease progression. Investigating cellular composition within tissue is fundamental for advancing biomedical research, revealing underlying mechanisms of disease progression such as tumorigenesis and tumor-TME (tumor microenvironment) interactions. The emerging use of single-cell RNA sequencing (scRNA-seq) has revolutionized our understanding of cellular heterogeneity by enhancing the genome-wide expression profile resolution from bulk-level to single-cell-level, enabling the characterization of distinct cell type within a complex mixture. However, the ongoing debate about how to properly normalize scRNA-seq data to achieve a more accurate representation of biology remains a contentious topic. For clarity in the explanations that follow, we will refer to the total number of mRNA molecules within each individual cell as the "true" transcriptome size. Current scRNA-seq technologies can capture and represent the "true" transcriptome size using the final raw counts, which is further termed as "measured" transcriptome size, short as transcriptome size. It has been observed in multiple scRNA-seq datasets[2–4] that cells of a single type from one specimen typically exhibit roughly the same transcriptome size. However, it significantly varies, often by multiple folds, across different cell types[2]. Technology-derived effects can cause the "measured" transcriptome sizes to vary across identical cell types within the same sample. However, at the population level, the cells whose "true" transcriptome sizes are larger than those of other cells within the same sample are likely to produce higher "measured" values. If the "measured" transcriptome sizes of a cell type surpass those of another, then, it is highly likely that the first cell type's "true" transcriptome sizes also exceed those of the second. This difference mirrors inherent biological diversity.

The technology-derived effects could result from sample preparation, platform selection, and sequencing depth, which are

Department of Computational Biology, St. Jude Children's Research Hospital, Memphis, TN 38105, USA. [✉]e-mail: Jiyang.Yu@stjude.org

typically viewed as the primary causes of variation in transcriptome size of single cell type between different specimens[5,6]. The differences in transcriptome size pose a challenge when directly comparing gene expression levels within or across specimens[7]. To address this issue, current scRNA-seq analysis algorithms typically operate on the assumption that the transcriptome size of all cells stays constant, thus apply count per 10 K (**CP10K**) normalization to the raw counts, which eliminates the technology-derived effects and enables comparisons and analyses between cells[8–10]. As a side effect, this normalization approach also removes the variations in transcriptome size caused by real biological changes, leading to an uneven **scaling effect** to the expression levels of genes (enlarged or shrank). This scaling effect usually has minimal impact on cells of the same type as they exhibit similar transcriptome size. Therefore, the CP10K-normalized scRNA-seq data can still distinguish different cell types and enable accurate clustering for annotation[11–14]. Consequently, the CP10K approach is commonly employed as the default setting for popular scRNA-seq analysis toolkits including Seurat[8,9], Scanpy[10], and Harmony[15]. However, when it comes to comparing different cell types, this scaling effect can create substantial problems, such as obstacles in pinpointing specific differentially expressed genes (DEGs) for the cell types of interest. It is worth mentioning that other normalization approaches such as CPM (counts per million), TPM[16], SCnorm[17], and SCTransform[6] are also susceptible to this scaling effect.

Though bulk RNA-seq does not produce an expression profile for each individual cell, it still has irreplaceable advantages over scRNA-seq, including (i) lower cost and less dependency on equipment, (ii) better quality with high gene coverage and low sparsity, and (iii) greater accessibility to fixed specimens collected from routine clinical care. Furthermore, an enormous amount of bulk RNA-seq data that cover all types of tissue and organs has been profiled and made available in the last two decades. Accurately and robustly parsing the proportions of each cell type contained in bulk samples enables the extraction of additional information from a large amount of existing bulk datasets, greatly enhancing their values. The growing quantity and improved precision of scRNA-seq data significantly enhance the deconvolution accuracy when used as references. Several recent algorithms including BayesPrism[18], CIBERSORTx[19], and MuSiC[20] are widely used and appreciated in the community. However, current deconvolution approaches face several significant challenges[21] arising from the variation of transcriptome size across cell types, technology differences, and biological noises. Due to the inherent differences of data generation between bulk RNA-seq and scRNA-seq, three types of issues occur when directly utilizing CP10K-normalized scRNA-seq data as reference for bulk RNA-seq data deconvolution: (i) the scaling effect caused by transcriptome size, (ii) the gene length effect caused by sequencing technique, and (iii) the expression variance size. None of the current methods are able to address these issues, resulting in biased deconvolution outputs that affect downstream analyses.

Here we present ReDeconv, a toolkit for scRNA-seq data normalization and bulk RNA-seq cellular deconvolution. It includes an innovative normalization approach specifically designed for scRNA-seq data, effectively preserving the influences of biological cell-to-cell variability while removing the one of technology-derived effects on transcriptome size of different cell types. We have conducted meticulous mathematical examinations of key issues encountered in the deconvolution of bulk RNA-seq data and systematic evaluations using synthetic and real datasets. ReDeconv addresses these challenges with its unique modeling, resulting in more reliable and precise outcomes in cellular decomposition.

## Results
### ReDeconv framework
To address the abovementioned challenges in cellular deconvolution analysis, we develop a comprehensive ReDeconv framework.

ReDeconv primarily aims to improve the scRNA-seq data analysis workflow (Fig. 1a) by providing a transcriptome-size-corrected normalization algorithm (Fig. 1b) and improve the accuracy of bulk RNA-seq cellular deconvolution by addressing crucial issues impacting deconvolution (Fig. 1c). Characterizing the inherent differences between bulk RNA-seq and scRNA-seq data has allowed us to identify three types of issues that are not addressed by existing methods.

**Type-I issues** occur when applying CP10K-normalized scRNA-seq data as a reference for bulk RNA-seq deconvolution (Fig. 1a). The scaling effects introduced by CP10K-normalized scRNA-seq reference can significantly impact the cellular deconvolution results, especially for rare cell types in TME. Unfortunately, CP10K normalization enlarges the smaller transcriptomes sizes of these infrequent cell populations, causing their proportion within the bulk sample to be underestimated and substantial discrepancies to be introduced into the final cellular deconvolution results.

**Type-II issues** are the gene length effects during library preparation for bulk RNA-seq samples. At present, most bulk RNA-seq is performed using the total RNA-seq protocol, whose raw counts are correlated with gene length[22]. TPM or RPKM/FPKM normalization is typically employed to mitigate this influence. In contrast, unique molecular identifier (UMI)-based scRNA-seq data[23] is not affected by gene length (Supplementary Fig. 1). The unmatched gene length effect on bulk and single-cell RNA-seq data impacts cellular deconvolution. For example, applying the same normalization approach (whether CP10K, CPM, TPM, or RPKM/FPKM) to both bulk and scRNA-seq datasets introduces this type-II issues into the deconvolution process.

**Type-III issues** emerge when the expression difference of genes within the same cell type in the reference and mixture samples is overlooked. Since the expression of each gene in each cell of a given type follows a particular distribution[24], the average expression of a gene for a specific cell type typically differs between the reference and mixture samples. scRNA-seq offers whole transcriptome profiling for all cells, providing information of expression distribution of each gene in each single cell-type. However, current deconvolution methods[18–20,25] typically utilize the mean value of gene expression for each cell type during reference construction and either do not consider the expression variances or set the variance to a constant, discarding information that could potentially improve their accuracy and robustness.

ReDeconv aims to circumvent these common issues (Fig. 1c). It addresses Type-I issues by normalizing the reference scRNA-seq data using Count based on Linearized Transcriptome Size (**CLTS**), an approach that preserves variations in the transcriptome sizes of different types of cells in scRNA-seq data. ReDeconv addresses Type-II issues by selectively applying TPM or RPKM/FPKM normalization to only bulk RNA-seq data. Finally, it addresses Type-III issues by integrating expression variance information into the deconvolution modeling, selecting only the signature genes whose expression is stable (fewer variables) within each type of cells for scRNA-seq reference construction (Methods).

### ReDeconv incorporates transcriptome size for scRNA-seq normalization
To confirm if the differences in transcriptome size across cell types are inherent to the biological effects, we analyzed and compared the Allen Institute's comprehensive single-cell atlas of the mouse whole cortex & hippocampus and human primary motor cortex datasets[4,26,27]. From 55 mouse specimens, we selected two mouse specimens (410107 as sample I, 446701 as sample II) and checked the transcriptome size across cell types (Supplementary Fig. 2a). Consistent with previous reports, we observed that, within a single sample, the transcriptome size will vary significantly across different cell types, but cells of the same type will have transcriptomes that are similar in size (Supplementary Fig. 2a). Furthermore, variations in transcriptome size of one

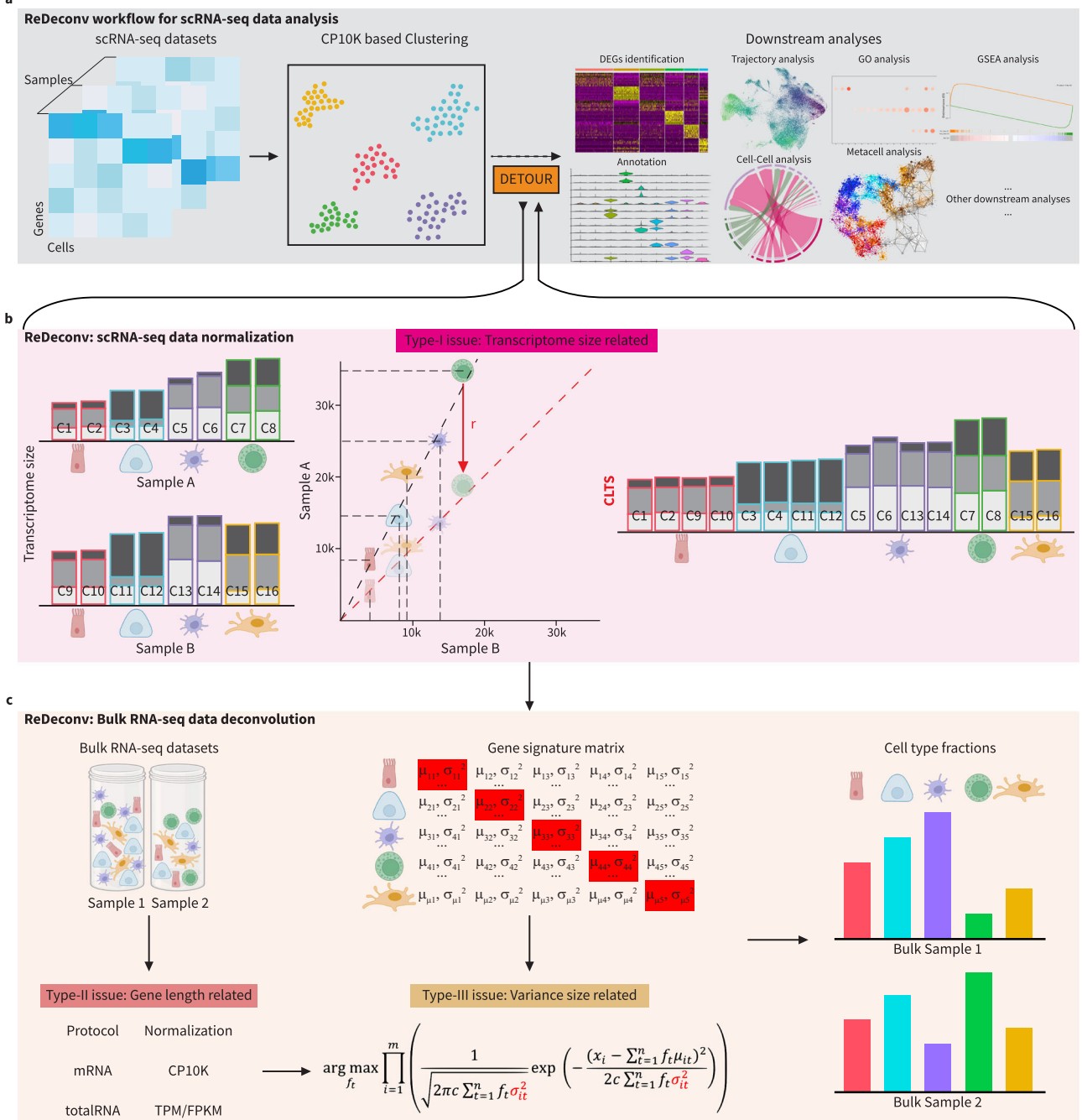

**Fig. 1 | A comprehensive view of the ReDeconv framework. a** ReDeconv-incorporated workflow for scRNA-seq data analysis. **b** Process of CLTS-normalization for scRNA-seq data to address transcriptome size-related (Type-I) issues. **c** The overall scheme of ReDeconv algorithm for bulk RNA-seq deconvolution with CLTS-normalized scRNA-seq reference. (The images of cells were created in BioRender)[54].

cell type among different specimens were also observed (Supplementary Fig. 2a). For instance, the average transcriptome size of L5 PT CTX cells in mouse sample I is approximately 21.6k, whereas it increases to 31.9k in mouse sample II (Fig. 2a). Given that the technology-derived effects would affect all cells within a sample equally, we hypothesized that the mean transcriptome sizes of the same cell type across any two samples would display a strong linear correlation. Indeed, when visualized the mean transcriptome sizes of different cell types across two mouse samples or lung cancer samples through a scatter plot (Fig. 2a and Supplementary Fig. 3) and heatmaps of Pearson correlation coefficient (Supplementary Fig. 2b, c), a clear and strong linear correlation was exhibited. Intriguingly, this

correlation not only persisted across species, as observed between mouse and human (Fig. 2b), but also held across different scRNA-seq methods[28] (Supplementary Fig. 4). These positive correlations between any two given samples strongly supported our hypothesis and indicated that the relative differences in transcriptome sizes among different cell types within one sample are highly stable, and the technology-derived effects do almost proportionally amplify the transcriptome sizes of all cells in the same sample. These observations suggested the inherent biological traits: cells of the same type demonstrate considerable similarity in (true) transcriptome sizes. On the other hand, significant disparities were observed between different cell types, which reflected their distinct expression profiles.

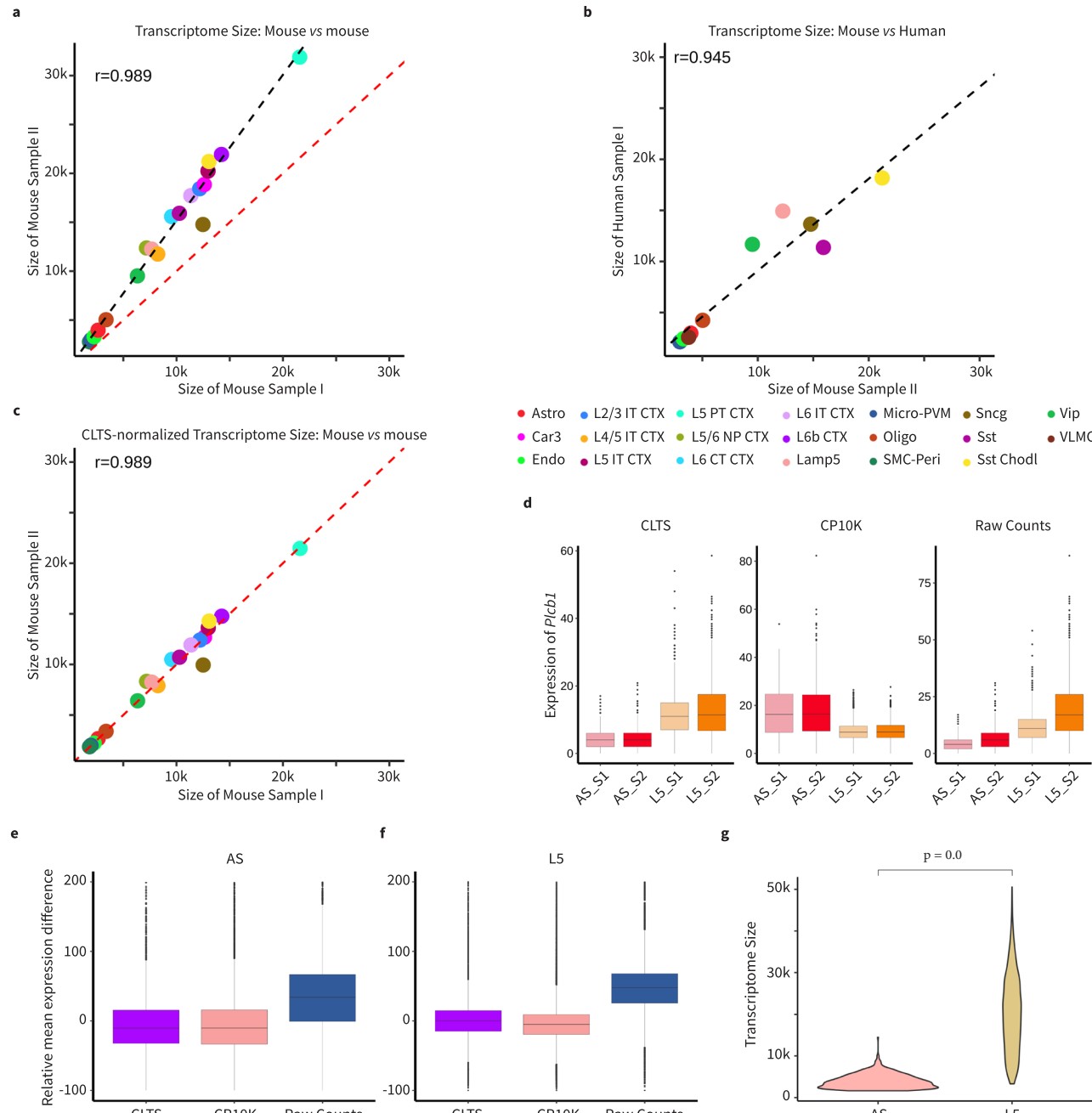

**Fig. 2 | Variation of transcriptome size and CLTS normalization. a–c** Scatter plots depict the comparison of mean transcriptome sizes of various cell types in two samples, demonstrating a strong linear correlation between the mean transcriptome sizes in the two samples. **a** Under raw count scRNA-seq data; mouse brain Sample_I vs. Sample_II. **b** Under raw count scRNA-seq data; mouse brain Sample_II (x-axis) vs. human brain sample (y-axis). **c** Under CLTS-normalized scRNA-seq data; mouse brain Sample_I vs. Sample_II. **d** Expressions of gene *Plcb1* in L5 and AS of mouse brain Sample_I (with 206 AS and 1665 L5 cells) and Sample_II (with 461 AS and 1857 L5 cells) under CLTS-, CP10K-normalized, and raw count scRNA-seq data, respectively. **e, f** Relative differences of all genes between mouse brain Sample-I and Sample_II for AS (**e**) or L5 (**f**) under CLTS-, CP10K-normalized, and raw count scRNA-seq data, respectively (only represented between −100 and 200). **g** Transcriptome sizes of mouse brain AS (*n* = 461) and L5 (*n* = 1857). In the box plots presented in this figure, the values are depicted as the median, represented by the middle line, and the 25th and 75th percentiles, represented by the box. In the violin plots showcased in this figure, the *P*-values were computed using a two-tailed t-test. Source data are provided as a Source Data file.

Given the intrinsic variation of transcription size across cell types, the widely used CP10K normalization introduces the scaling effects to different cell types at varying ratios (Supplementary Fig. 2d). To address this issue, we present **CLTS** (Counts based-on Linearized Transcriptome Size), a normalization approach for scRNA-seq data that not only preserves the transcriptome size information for different cell types within the same sample, but also mitigates the technology-derived effects across different samples (Fig. 2c,

Supplementary Fig. 2e, Methods). Specifically, when the scRNA-seq data is generated from a single sample, CLTS would consider that the transcriptome size directly reflects the biological differences in total mRNA expression and therefore utilize the raw counts for downstream analyses. When the scRNA-seq data is generated from two or more samples, CLTS primarily relies on the linear relation of average transcriptome sizes for the shared cell type across samples to perform normalization. Consequently, after normalization, the same cell types

maintain a similar transcriptome size mean across all samples, while different cell types retain their transcriptome size diversity, resulting in the CLTS normalized scRNA-seq data for downstream analyses.

## ReDeconv corrects CP10K-misidentified differentially expressed genes

An effective normalization method for single-cell data would preserve the gene expression variability within each cell, maintain transcriptional distinctions among the same cell type, and accurately reflect differences across various cell types. To demonstrate the superiority of CLTS normalization on scRNA-seq downstream analyses, we first examined the influence of CLTS and CP10K normalizations on the gene expression in astrocytes (**AS**) and L5 IT CTX neuronal cells (**L5**) in mouse brain samples I and II. We observed that the technology-derived effects amplified the raw count gene expression in samples I and II disproportionately. This resulted in higher gene expression in both AS and L5 cells in sample II compared to sample I (Fig. 2d, Supplementary Fig. 5a–c). The huge positive bias of relative expression differences, defined by the equation $\frac{\exp_{sample\_II} - \exp_{sample\_I}}{\exp_{sample\_I}} *100$, for all genes in both AS and L5 cells, elucidated that the influence of these technology-derived effects on the gene expression in samples I and II is universal (Fig. 2e, f, Supplementary Fig. 5d, e). Both CP10K and CLTS normalizations effectively mitigated these technology-derived effects, equalizing gene expression levels within identical cell types across both samples. Consequently, the relative expression differences for all genes in both AS and L5 cells between samples I and II were significantly reduced, eliminating their substantial positive bias (Fig. 2e, f, Supplementary Fig. 5d, e). As discussed above, the primary issue with CP10K approach is the scaling effects on gene expression of different cell types within the same samples, rendering gene expression incomparable. For example, genes *Plcb1, Ntm, Cpe*, and *mt-Atp6* exhibited higher raw counts in L5 compared to AS. However, only CLTS-normalized data preserved these trends, whereas the CP10K-normalized data exhibited reversed results, higher expression of these genes in AS compared to L5 (Fig. 2d, Supplementary Fig. 5a–c). This occurred because the CP10K approach tended to enlarge the overall gene expression in cells with smaller transcriptome sizes, while shrinking those in cells with larger transcriptome sizes. In this case, the average transcriptome size of L5 was approximately five times greater than that of AS, leading to a significant shrinking effect to gene expression in L5 (Fig. 2g).

To further validate the differential expression results of scRNA-seq data normalized by CLTS and CP10K, we used the corresponding CosMx data of mouse brains as the ground truth[29]. For 940 shared genes in scRNA-seq and CosMx datasets, we calculated the log2 fold-change of expression means between L5 and AS. A positive value indicated upregulation of the gene in L5, whereas a negative value indicated downregulation. Our results indicated that 86.7% (81.9% positive + 4.8% negative) of genes from CLTS-normalized data kept the same fold-change directions with those in the CosMx data (Fig. 3a). In contrast, only 62.4% (56.7% positive + 5.7% negative) of genes showed this consistency between the CP10K-normalized and CosMx data (Fig. 3b). It is worth mentioning that CP10K approach relatively shrank the overall gene expression in L5, which resulted in a downward shift of the scatter plot. This shift resulted in changing the fold-change directions for 24.4% of genes, causing their directions to disagree with the CosMx data.

We further employed a log2 fold change of 1.5 and a p-value of 0.05 (determined by the Wilcoxon rank-sum test) as the cutoff to select *significantly* down-regulated genes of L5 from CosMx, CP10K-, and CLTS-normalized data, respectively (Fig. 3c). Our particular focus was on the 107 *significantly* down-regulated DEGs, which were identified by CP10K-normalized data, but were not detected by CLTS-normalized data. Intriguingly, none of these genes were significantly down-regulated in the CosMx data. Furthermore, 93.5% (83.2% + 10.3%)

of these genes were actually up-regulated in L5 according to the CosMx data, making it impossible to be down-regulated even with adjusted thresholds (Fig. 3d). Thus, the majority of 107 genes identified as significantly down-regulated in L5 IT cells under only CP10K-normalized data were not corroborated by the CosMx data.

Similarly, among the 259 *significantly* up-regulated DEGs identified by CLTS-normalized data but not by CP10K-normalized data, 98.5% (34.4% + 64.1%) of them showed consistent up-regulated trends, and 61.4% of them were identified as significantly up-regulated DEGs in CosMx data (Fig. 3e, f). On the other hand, 65.3% (1.2% + 64.1%) of these genes exhibited opposite down-regulated trends and 29.7% were misidentified as significantly down-regulated DEGs by CP10K (Fig. 3f). To further validate the accuracy of DEG identification, we checked the expression of misidentified DEGs, including *Plcb1, Tnik*, and *Rora* in both L5 and AS, as well as the actual gene signals in CosMx images (Fig. 3g, h). These genes also demonstrated significant enrichment in neuro-associated pathways (Supplementary Fig. 6a), consistent with the neuron nature of L5. These results confirmed that *Plcb1, Tnik*, and *Rora* had higher expression in L5 compared to AS, and CLTS-normalized data reflected the ground truth faithfully while CP10K-normalized data displayed opposite results.

In addition the mouse data, we further evaluated ReDeconv's ability to correctly identify DEGs using published scRNA-seq[27] and CosMx data[30] of human brain samples. We focused on L2/3 IT CTX neuronal cells (**L2/3**) and astrocytes (**AS**). Similarly, the average transcriptome size of L2/3 was approximately five times greater than that of AS (Supplementary Fig. 7a). The results aligned well with our prior observations in mouse brains. (i) For 5,746 shared genes in L2/3 in scRNA-seq and CosMx datasets, the CosMx data showed a greater level of consistency (in terms of fold change direction) to the CLTS-normalized data (92.2%) compared to the CP10K-normalized data (59.9%, Supplementary Fig. 7b, c). (ii) For 1,027 significantly down-regulated DEGs in L2/3 identified under CP10K-normalized but not CBTS-normalized data, only 5.6% was supported by CosMx data (Supplementary Fig. 7d, e). (iii) For 2,559 significantly up-regulated DEGs in L2/3 identified under CLTS-normalized but not CP10K-normalized data, 77.7% was further confirmed by CosMx data (Supplementary Fig. 7f, g). (iv) CLTS-normalized data faithfully reflected the expression levels of selected genes in L2/3 and AS, while CP10K-normalized data displayed opposite results (Supplementary Fig. 7h, i). Functional enrichment analysis using these genes further supported our conclusion (Supplementary Fig. 6b).

Previously, scRNA-seq data with cells from two mouse samples was used to evaluate the performance of CLTS. Incorporating the scRNA-seq data of cells from additional mouse samples also demonstrated that CLTS normalization reduced variation in the gene expression of a given cell type across samples. Furthermore, comparing the gene expression of L5 and Astro cells revealed significant overlap between DEGs (up-regulated in L5 cells) obtained from four-sample and two-sample CLTS-normalized scRNA-seq data, respectively (Supplementary Fig. 8).

Collectively, these results demonstrated that the CLTS normalization in ReDeconv outperformed the commonly used CP10K approach by retaining the effects of biological distinctions while mitigating those resulting from technology-derived effects on the transcriptome size of various cell types. ReDeconv serves as a valuable addition to the conventional workflow for scRNA-seq data analysis (Fig. 1a). Following the completion of clustering, an extra step of CLTS normalization will improve DEG or marker identification, and possibly other downstream analyses (Fig. 1b).

## ReDeconv provides accurate bulk cellular deconvolution in synthetic data

To benchmark the overall performance of ReDeconv with three popular bulk RNA-seq deconvolution algorithms, including BayesPrism,

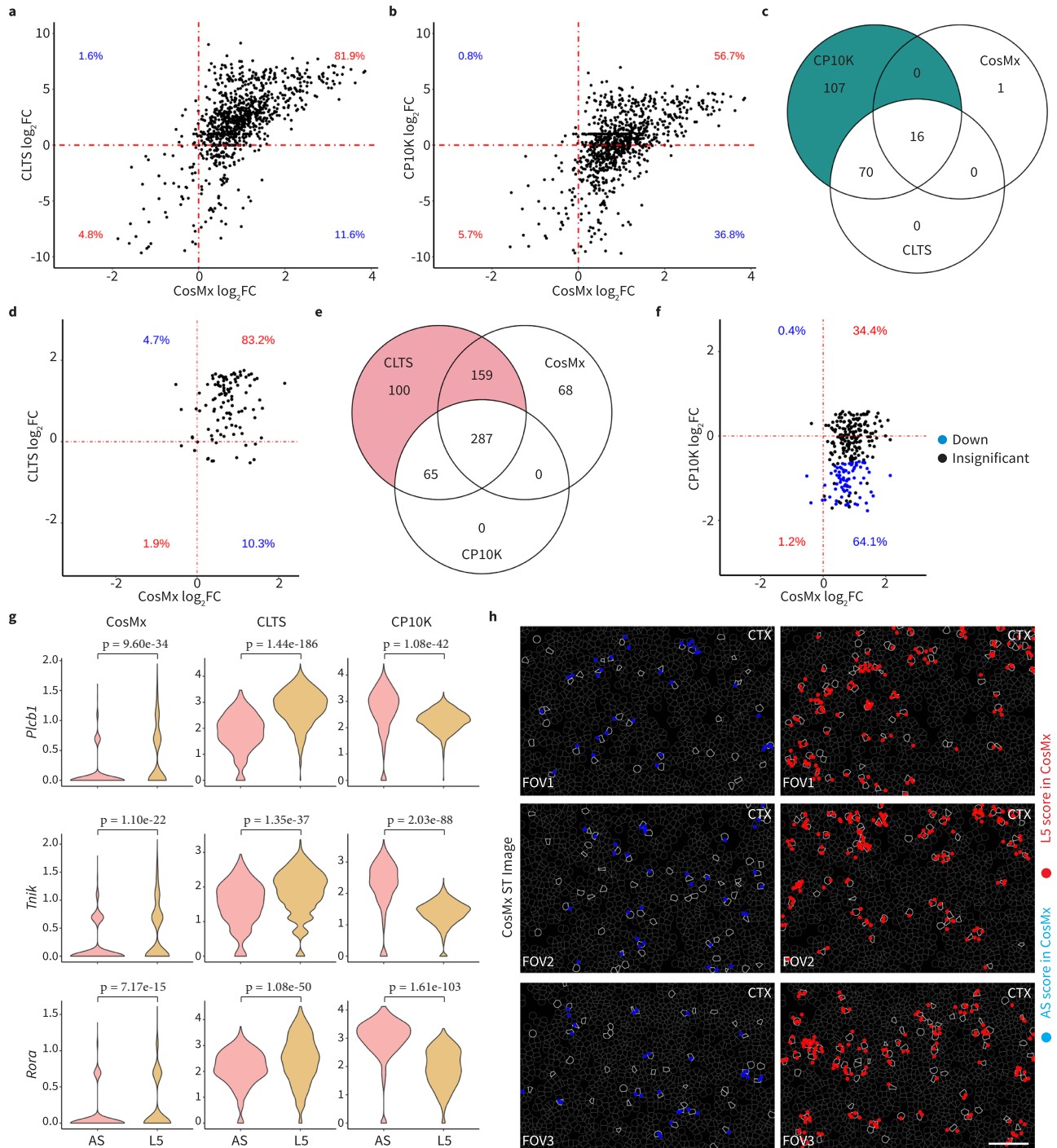

**Fig. 3 | CLTS corrects CP10K-misidentified differentially expressed genes from scRNA-seq data of mouse brains using CosMx data as ground truth. a, b** Scatter plots comparing the fold-changes of genes between L5 and AS in CosMx data with those in scRNA-seq data under CLTS-normalization (**a**) or CP10K-normalization (**b**). **c** Overlap of significantly down-regulated genes in L5 (vs. AS) under CosMx data, CLTS-normalized and CP10K-normalized scRNA-seq data. **d** Scatter plots comparing the fold-changes of genes in CosMx data with those in scRNA-seq data under CLTS-normalization, where these genes were identified as significantly down-regulated in L5 (vs. AS) under CP10K-normalized but not CLTS-normalized scRNA-seq data. **e** Overlap of significantly up-regulated genes in L5 (vs. AS) under CosMx data, CLTS-normalized and CP10K-normalized scRNA-seq data. **f** Scatter plots comparing the fold-changes of genes in CosMs data with those in scRNA-seq data under CP10K-normalization, where these genes were identified as significantly up-

regulated in L5 (vs. AS) under CLTS-normalized but not CP10K-normalized scRNA-seq data. **g** Expression of *Plcb1*, *Tnik*, and *Rora* in AS and L5 under CosMx data, CLTS-normalized, and CP10K-normalized scRNA-seq data, respectively. **h** The actual gene signals in CosMx images for genes in (**g**). In the images, the white circles in three FOVs (on the left) represent AS cells, and the blue dots represent the expression of *Plcb1*, *Tnik*, and *Rora* (from top to bottom). Similarly, the white circles in the same three FOVs (on the right) represent L5 cells, and the red dots represent the expression of (from top to bottom) *Plcb1*, *Tnik*, and *Rora*. All three genes exhibited greater expression in L5 cells than in AS cells, consistent with the CLTS approach. In the violin plots showcased in this figure, the P-values were computed using a two-tailed t-test. CosMx: AS ($n = 993$), L5 ($n = 651$); CLTS and CP10K: AS ($n = 461$), L5 ($n = 1857$). Source data are provided as a Source Data file.

CIBERSORTx, and MuSiC, we constructed a total of six synthetic bulk RNA-seq datasets using different scRNA-seq sources, including human brain samples[27], GBM samples[18], blood samples[19], and lung cancer samples[31], named as synthetic bulk datasets (SYN Data A-F, Methods). Following the instructions in the methods' manuals, for all tests, we used raw count scRNA-seq and TPM bulk RNA-seq data for ReDeconv. Meanwhile, the other three methods were tested using raw count scRNA-seq and bulk RNA-seq data. Consequently, these three benchmarking methods encountered Type-II and Type-III issues, with BayesPrism and CIBERSORTx further facing Type-I issues. Conversely, ReDeconv didn't exhibit any of these three types of issues. In SYN Data A from human brain samples[27], six abundant cell types (L5 ET, L6b, L5/6 NP, Lamp5 neuronal cells, oligodendrocyte, astrocyte) were mixed with equal proportions (Methods). Predicted fractions and relative error, $\frac{proportion_{prediction}-proportion_{ground\_truth}}{proportition_{ground\_truth}}*100$, were used to evaluate the inferred cell proportions by four methods (Fig. 4a, Supplementary Fig. 9a). ReDeconv's outputs were astoundingly accurate with a mean proportion for each cell type equal to the ground truth ($1/6 = 0.167$) and a maximal standard deviation of $5.43 \times 10^{-3}$. The outputs also maintained the lowest relative error while those from the other three methods exhibited significant deviation from the ground truth. Particularly, BayesPrism predicted an average proportion of L5 ET neuron that was over 0.404, 2.4 times greater than the ground truth. Correspondingly, BayesPrism's outputs displayed an average relative error of 142.4% for L5 ET neuron. CIBERSORTx also overestimated the proportion of L6b neurons with an average greater than 0.325 (with a mean relative error of 94.9%). MuSiC, on the other hand, underestimated astrocyte's abundance to less than 0.021 (with an average relative error of -87.1%).

Similar results were observed when using SYN Data B-D with equal cell proportion mix, further supporting the superiority of ReDeconv. ReDeconv consistently produced results that are significantly closer to the ground truth, regardless of the total number of cell types involved. In contrast, the cell type proportions predicted by the other three methods significantly deviated from the ground truth (Supplementary Fig. 9b–d).

To rule out that the superiority of ReDeconv was equal-proportion synthetic bulk data dependent, we then generated two additional SYN Data E and F with uneven cell-type proportions (Fig. 4b, c, Methods). In SYN Data E, all 100 mixed samples possessed the same fraction for each cell type, with the proportions of different cell types being randomly generated (Fig. 4b, Supplementary Fig. 10a). Conversely, in SYN Data F, the proportions of varied cell types were randomly generated for each of the 100 mixed samples individually (Fig. 4c, Supplementary Fig. 10b). In both two datasets, the results aligned closely with those from SYN Data A-D. The predictions of ReDeconv were remarkably close to the ground truth, exhibiting minimal relative errors and standard deviations. On the contrary, predictions from the other three methods significantly deviated from the ground truth, resulting in substantial relative errors. Altogether, the experiments reported herein supported the accuracy and robustness of cellular composition inferred by ReDeconv.

### ReDeconv outperforms other deconvolution methods in real data

Besides surpassing its peers using synthetic datasets, ReDeconv maintained an advantage when applied to real bulk RNA-seq data. For this evaluation, we first used a publicly available bulk RNA-seq data[32] with 18 mixture samples each of which contained varying quantities of cells from six cell lines. The precise composition determined by actual cell counts was used as the ground truth. The matched scRNA-seq data mixed with six cell lines was used as a reference. Evaluated by Pearson's correlation coefficient of predicted cell-type proportions against the

ground truth (Fig. 4d), ReDeconv continued to perform the highest correlation (r = 0.963), followed by CIBERSORTx with decent performance (r = 0.915), and BayesPrism (r = 0.728) and MuSiC (r = 0.413). To further assess ReDeconv's cellular deconvolution in comparison to other methods, we also calculated the relative prediction error. The predictions from BayesPrism and MuSiC yielded higher relative errors for abundant (>20%) cell types while those from CIBERSORTx and MuSiC exhibited significantly higher relative errors for rare cell types (≤20%). However, ReDeconv achieved the best performance for the predictions of both the rare and abundant cell types (Supplementary Fig. 10c).

The superiority of ReDeconv was also apparent in a second real bulk RNA-seq dataset[33]. In this dataset, each bulk RNA-seq sample primarily comprised one cell type, which was predominantly derived from sorted cells. Thus, the dominant cell type was known, although the exact proportion value was missing in each bulk sample. The evaluation was to check if the inferred dominant cell type by the algorithm matched the expected one. ReDeconv outperformed its competitors in capturing the dominant cell type in 4 out of 5 bulk samples, with the extremely low proportions of other cell types in its results (Fig. 4e). The only exception occurred because CD8$^+$ T cells were partially mis-identified as CD4$^+$ T cells, whose gene expression profiles shared high similarity. Conversely, other methods' performance is far from expectations. For CD4$^+$ T cell-dominant bulk sample, MuSiC's estimation of CD4$^+$ T cell proportion was near 0, while BayesPrism's estimation was also underestimated at less than 0.2. As for the NK cell-dominant bulk samples, CIBERSORTx failed to distinguish NK cell from CD8$^+$ T cell, monocyte, or NKT cell, leading to similar proportions to these four cell types.

Altogether, the benchmarking reported herein supported the accuracy and robustness of cell-type proportions generated by ReDeconv, which were biologically meaningful, and suitable for using with a variety of computational packages for downstream analysis. It also indicated that the three types of issues discussed indeed had significant impacts to cellular deconvolution processes. These findings demonstrated the superiority and validity of ReDeconv by addressing these issues.

### Impacts of Type-I and Type-II issues on bulk deconvolution

To further assess the impacts of Type-I and Type-II issues on cellular deconvolution, we introduced these issues into ReDeconv by inputting different formats of bulk and scRNA-seq data pairs. We also manipulated the influence of Type-I and Type-II issues on BayesPrism, CIBERSORTx, and MuSiC by utilizing the format of scRNA-seq and bulk RNA-seq data that differed from the guidelines provided in their respective manuals. The SYN Data A-D were used for this evaluation, where the transcriptome sizes among different cell types exhibited significant differences. When using CP10K-normalized scRNA-seq data paired with TPM/FPKM-normalized bulk RNA-seq data, all methods had Type-I issues and were free from Type-II issues. Intriguingly, all results exhibited remarkably similar and consistently biased deviations, indicating that the influences of Type-I issues were significant and algorithm-independent. Furthermore, the bias demonstrated a strong positive correlation with the transcriptome sizes of cell types (Fig. 5a, Supplementary Fig. 11). Specifically, all methods tended to overestimate the proportions of cell types with large transcriptome size, while underestimating those with small ones. These results aligned with our mathematical analysis on the influence of Type-I issues on deconvolution (Methods). From Eq. (2) in the Methods section, it is obvious that Type-I issues significantly influence the deconvolution results when the proportions of each cell type are not insignificantly small. Therefore, for these synthetic data, the Type-I issues predominantly affect the results of BayesPrism and CIBERSORTx, even though they also suffer from Type-III issues.

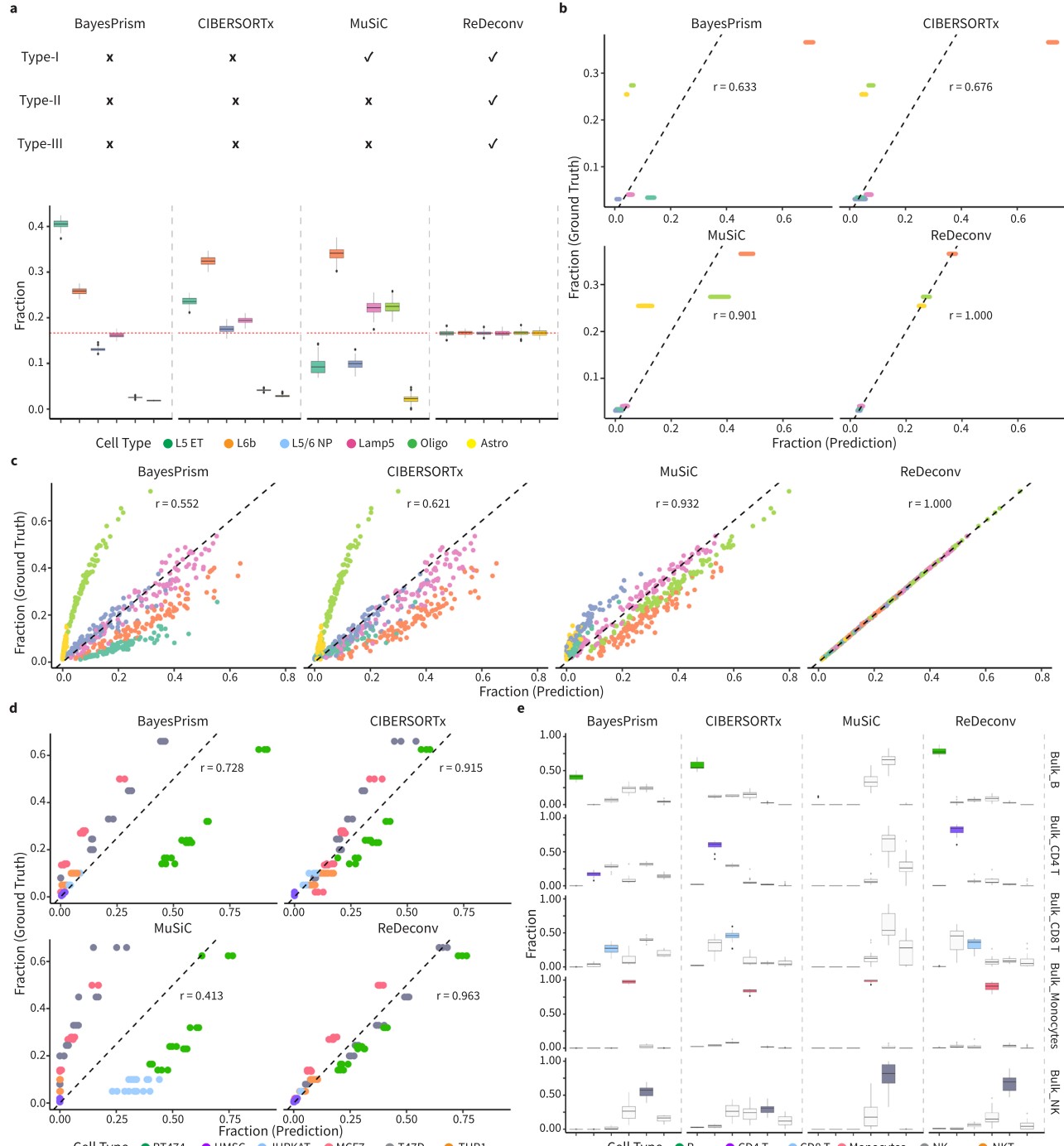

**Fig. 4 | Overall performance of ReDeconv vs. popular bulk deconvolution methods evaluated using synthetic and real datasets. a** Cell type fractions predicted by BayesPrism, CIBERSORTx, MuSiC, and ReDeconv against ground truth using bulk RNA-seq data and corresponding scRNA-seq data. Recomended input formats and parameters were used according to each method's manual. Bulk RNA-seq data were 100 synthetic mixture samples with equal fractions to all cell types (SYN Data A). In all tests, CIBERSORTx and BayesPrism exhibited all three types of issues, while MuSiC showed Type-II and Type-III issues. ReDeconv, however, didn't exhibit these three types of issues. **b, c** Correlation of predicted fractions with ground truth using 100 synthetic mixture samples with identical fractions for each fixed cell type, such that the fractions of different cell types are randomly generated (SYN Data E) (**b**) and 100 synthetic mixture samples with fractions of all cell types for each sample were randomly generated (SYN Data F) (**c**). **d** Evaluation using true bulk RNA-seq data of mixture samples using 6 cell lines with known fractions for each cell line. **e** Evaluation using true bulk RNA-seq data of sorted PBMC B, CD4 T, monocytes, NK, and NKT cells (*n* = 17 for each cell type of sorted bulk samples). In the box plots presented in this figure, the values are depicted as the median, represented by the middle line, and the 25th and 75th percentiles, represented by the box. Source data are provided as a Source Data file.

---

Not all existing methods suffered from both Type-I and Type-II issues. MuSiC could avoid Type-I issues when it used raw count scRNA-seq data as reference. However, MuSiC also recommended using raw counts of bulk RNA-seq data as input, leading to the occurrence of type-II issues (MuSiC in Fig. 4, Supplementary Figs. 9, 10). These results indicated that Type-II issues also significantly influenced the deconvolution. A comprehensive mathematical analysis detailing how Type-II issues impact convolution can be found in the Methods section. For

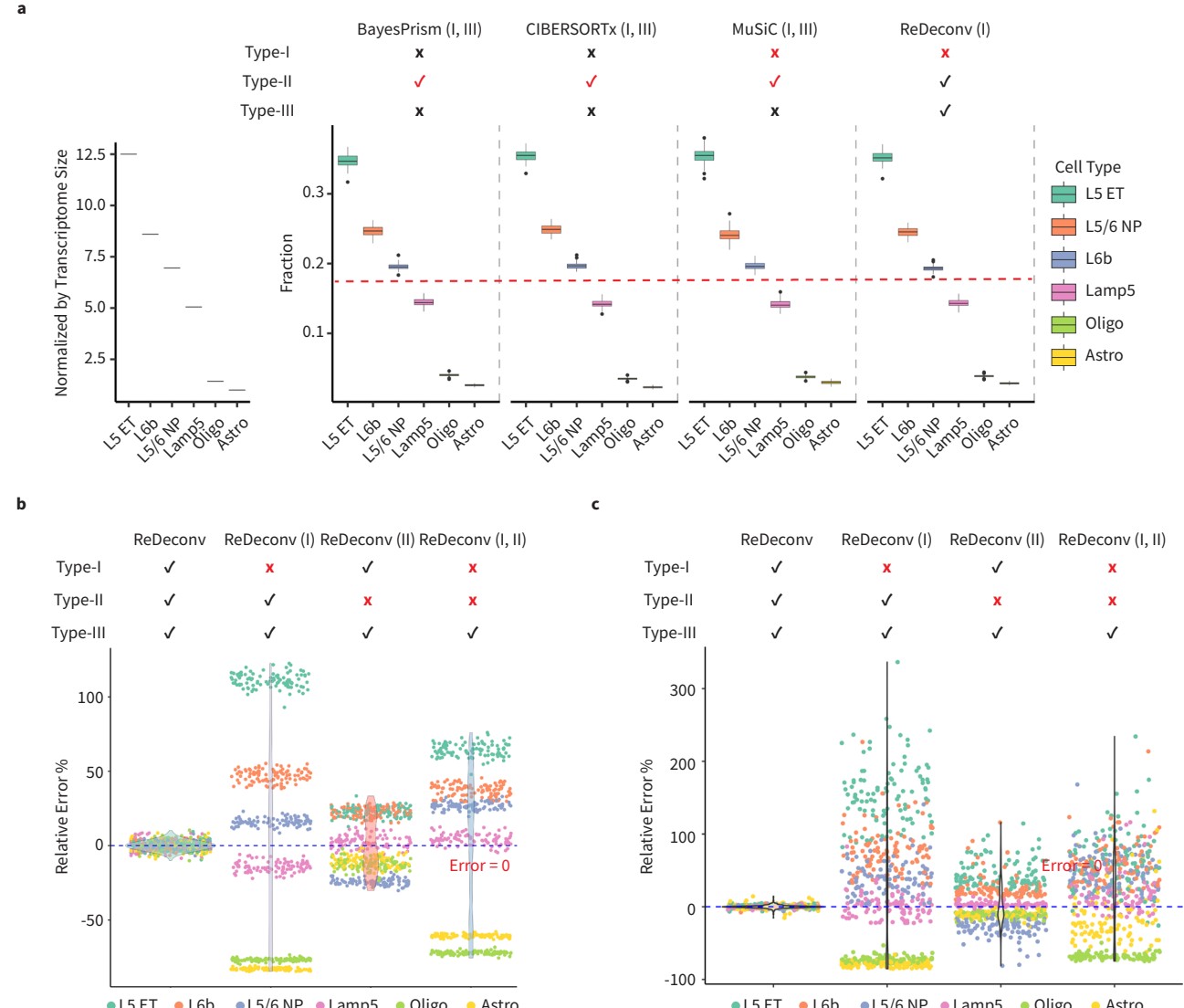

**Fig. 5 | Examining the impacts of Type-I and/or Type-II issues on deconvolution. a** Simulation results illustrating the impact of Type-I issue on the deconvolution output of four models, using synthetic bulk RNA-seq data of 100 samples with equal fractions for all cell types (SYN Data A). **b, c** Utilization of ReDeconv to demonstrate the effects of Type-I, Type-II issues, or both on the deconvolution results using 100 synthetic mixture samples with equal fractions to all cell types (SYN Data A) (**b**) and 100 synthetic mixture samples with randomly generated fractions (SYN Data F) (**c**). In this simulation, Type-I and/or Type-II issues were deliberately introduced to ReDeconv. In any given method, a red 'x' or '√' denotes modifications to the model, implying that recommended format from manuals for

inputs are not used. Consequently, a specific type of issue is either introduced (indicated by a red 'x') or resolved (indicated by a red '√'). For instance, in Fig. 5a, Type-I issues were incorporated into the modified versions of MuSiC and ReDeconv, whereas Type-II issues were resolved in the modified versions of BayesPrism, CIBERSORTx, and MuSiC. The term 'BayesPrism (I, III)' signifies that the adjusted version of BayesPrism continues to exhibit Type-I and III issues. In the box plots presented in this figure, the values are depicted as the median, represented by the middle line, and the 25th and 75th percentiles, represented by the box. Source data are provided as a Source Data file.

BayesPrism and CIBERSORTx, we noticed that CP10K normalization was applied to scRNA-seq reference automatically, therefore, Type-I issues were consistently present in both methods, regardless of what formats of bulk and scRNA-seq datasets used as inputs. Thus, the impacts of only Type-II issues on BayesPrism and CIBERSORTx couldn't be evaluated.

Next, we employed ReDeconv to probe the individual and additive impacts of Type-I and Type-II issues on deconvolution results, ensuring that the outcomes remained unaffected by Type-III issues. The relative errors were used for performance evaluation. The presence of Type-I issues led to a relative error exceeding 100%, whereas Type-II issues could trigger a relative error of over 110%. The simultaneous occurrence of both Type-I and Type-II issues resulted in a relative error of less than -80% (Fig. 5b, c, Supplementary Fig. 12). It was worth

mentioning that Type-I and Type-II issues did not necessarily have a cumulative effect. For instance, Type-II issues may reduce the proportion of a cell type with large transcriptome size, counterbalancing the scaling effects from Type-I issues. In this case, the relative error for a cell type impacted by both Type-I and Type-II issues could be more "in line" with the ground truth compared to one that is only influenced by the Type-I issues (Methods).

It is important to note that, in addition to CP10K, other normalization methods with scaling issues that can impact deconvolution when applied to reference scRNA-seq data include scNorm and scTransform. These can lead to Type-I issues if utilized to normalize scRNA-seq data that is used as a reference for deconvolution (Supplementary Fig. 13).

## Addressing type-III issues improves deconvolution of rare cell types

To evaluate the improvement of ReDeconv by addressing Type-III issues, we applied both ReDeconv and MuSiC with TPM-normalized SYN Data E-I and the associated raw count scRNA-seq data, which ensured that MuSiC remained unaffected by Type-I and Type-II issues. Since we could not remove the Type-I issues from CIBERSORTx and BayesPrism by changing the normalization approach (no access to the source code), they were therefore excluded in this benchmarking. In these assessments, ReDeconv addressed Type-III issues, but MuSiC didn't (Fig. 6a, b, Supplementary Fig. 14). Both methods produced cell-type proportions that were reasonable and close to the ground truth, with a Pearson's correlation coefficient of over 0.999. A further comparison of the relative error of the results revealed that (i) the relative error for both methods tended to increase for rare cell populations, however, ReDeconv demonstrated smaller deviations from zero; (ii) the relative errors generated by ReDeconv exhibited smaller variances than those by MuSiC and the variances were significantly smaller (with p-values from F-test to be <0.05) in most of cases (Supplementary Table 1). In a further evaluation using a real bulk RNA-seq data[32], similar outcomes were observed where ReDeconv exhibited substantially smaller relative errors than those seen with MuSiC, particularly for rare cell populations (ReDeconv in Fig. 4d; Supplementary Fig. 15).

To facilitate a fair assessment of the impact of Type-III issues on CIBERSORTx and BayesPrism, which consistently exhibited Type-I issues, we took advantage of two sorted bulk RNA-seq datasets for these evaluations. In both datasets, each bulk RNA-seq sample is primarily comprised of one single cell type, which significantly reduced Type-I issues' impacts. We deliberately introduced Type-I issues to ReDeconv to illustrate that their impact on deconvolution is minimal in these sorted bulk RNA-seq data (Fig. 6c, Supplementary Fig. 16). For these tests, we used TPM-normalized bulk RNA-seq data and CLTS-normalized references as inputs. As a result, all models were free of Type-II issues, and the influence of Type-I issues on BayesPrism and CIBERSORTx is minimal. Therefore, the primary distinction between ReDeconv and the other methods lied in ReDeconv addressing Type-III issues, while the others do not.

Next, we evaluated the robustness of these methods using a fixed sorted bulk RNA-seq data including 5 cell types from PBMCs of COVID-19 viremia patients[33], pairing with different references. In test I, the identical sorted bulk RNA-seq data was also used as a reference. As expected, all methods performed quite well, with the exception of MuSiC, which had slightly poorer predictions (Supplementary Fig. 16a). In test II, we used a second sorted bulk RNA-seq data of PBMCs from Zika virus infected patients[34] as a reference. ReDeconv generated nearly identical and accurate cellular composition predictions regardless of the presence or absence of Type-I issues, indicating the minimal impact of Type-I issues on this bulk data. BayesPrism, CIBERSORTx, and MuSiC failed to identify the dominant cell type for at least one sorted bulk sample (Fig. 6c). In test III, scRNA-seq data of PBMCs from NSCLC patients[19] was used as a reference. There was a substantial worsening in the performance of BayesPrism, CIBERSORTx, and MuSiC; ReDeconv continued to perform well, however, with the caveat that it did mis-assign some CD8Ts to CD4Ts (Supplementary Fig. 16b). In test IV, scRNA-seq data of health PBMC samples[19] was used as reference. Again, ReDeconv outperformed other models, despite annotating some CD8Ts as CD8 NKT-like cells (Supplementary Fig. 16c). The results demonstrated that when utilizing different references for the same bulk RNA-seq sample deconvolution, the results from BayesPrism, CIBERSORTx, and MuSiC usually exhibited significant changes, while ReDeconv continued to be relatively stable, suggesting the importance of addressing Type-III issues. The fact that the performance of BayesPrism, CIBERSORTx, and MuSiC were improved when using health PBMC data (test IV) or the identical data (test I) as references indicated that a closely matched reference was a

circuital factor for good deconvolution performance. When applying the second sorted bulk RNA-seq datasets[35], all benchmarking methods provided decent results and managed to recognize the dominant cell types when using the same second sorted bulk RNA-seq or scRNA-seq data of the same tumor types as reference (Supplementary Fig. 16d, e). Taken together, these results demonstrated the significant impacts of type-III issues on the robustness and reliability of deconvolution. By addressing this, ReDeconv produced more stable cellular composition results and outperformed existing methods, even though they did not suffer from Type-I and Type-II issues.

## Discussion

We have introduced ReDeconv, an innovative method that offers a scRNA-seq normalization approach, CLTS, to address the transcriptome size-related issues (Type-I), benefiting conventional downstream analyses of scRNA-seq data including DEG identification and bulk RNA-seq deconvolution. A cell's transcriptome size is closely related to its cellular state and function. Significant differences in transcriptome sizes among different cell types have been observed across various specimens. These variations remain highly consistent and displayed a strong linear correlation across samples, even across species, indicating a reflection of real biological cell-to-cell variability. It is widely acknowledged that CP10K normalization makes the data comparable yet are not necessarily optimal. CLTS normalization has shown notable superiority over CP10K normalization. ReDeconv also identifies and resolves the Type-II issues encountered during bulk RNA-seq deconvolution, stemming from disparities in library preparation protocols between current UMI-based scRNA-seq and bulk total RNA-seq protocols. For the first time, ReDeconv introduces expression variance size for each cell type (Type-III issues) to the modeling to improve bulk deconvolution accuracy and robustness.

As mentioned in the main text, CP10K minimally impacts the clustering of scRNA-seq data, given that clustering takes into account the similarity of cell expression profiles. For methods that employ Euclidean distance to measure this similarity, it becomes crucial to implement CP10K/CPM to mitigate the impact of sequencing depth. However, if using scMINER[36], which gauges expression profile similarity through mutual information, we can bypass the need for CP10K/CPM in cell clustering. However, when assessing marker gene expressions to identify the cell type of each cluster, CP10K/CPM may lead to incorrect cell type annotations if certain marker genes exhibit substantial expressions in more than one cell type. Hence, we recommend integrating ReDeconv with tools like Seurat, Scanpy, and scMINER for cell-type annotations of scRNA-seq data. The demonstration codes for these integrations are available under Code Availability.

Both Type-I and Type-II issues have been frequently overlooked in several published biological studies with bulk RNA-seq deconvolution involved. For example, CP10K-normalized scRNA-seq data were often misused as a reference for deconvolution, leading to the occurrence of Type-I issues[37–48]. Type-I issues also impact the deconvolution of spot-level spatial transcriptomics data. A recent investigation explored the effect of various normalization strategies on scRNA-seq and bulk RNA-seq data and their impacts on some deconvolution models[32]. The study found that TPM-normalized bulk RNA-seq data paired with raw-count scRNA-seq data consistently outperformed other combinations. This is an expected outcome as these combinations would avoid both Type-I and Type-II issues, which is supported by our observations and evaluations. CP10K and SCTransform are the two normalization options for Seurat[8,9,49,50]. However, both approaches would cause scaling effects to gene expression of different cell types, introducing the Type-I issues and cause potential deviations to downstream analyses. For future reference, we have summarized deconvolution scenarios involving different input combinations of single cell and bulk RNA-seq data to serve as a reference for selecting normalization methods in future deconvolution analyses (Supplementary Table 2).

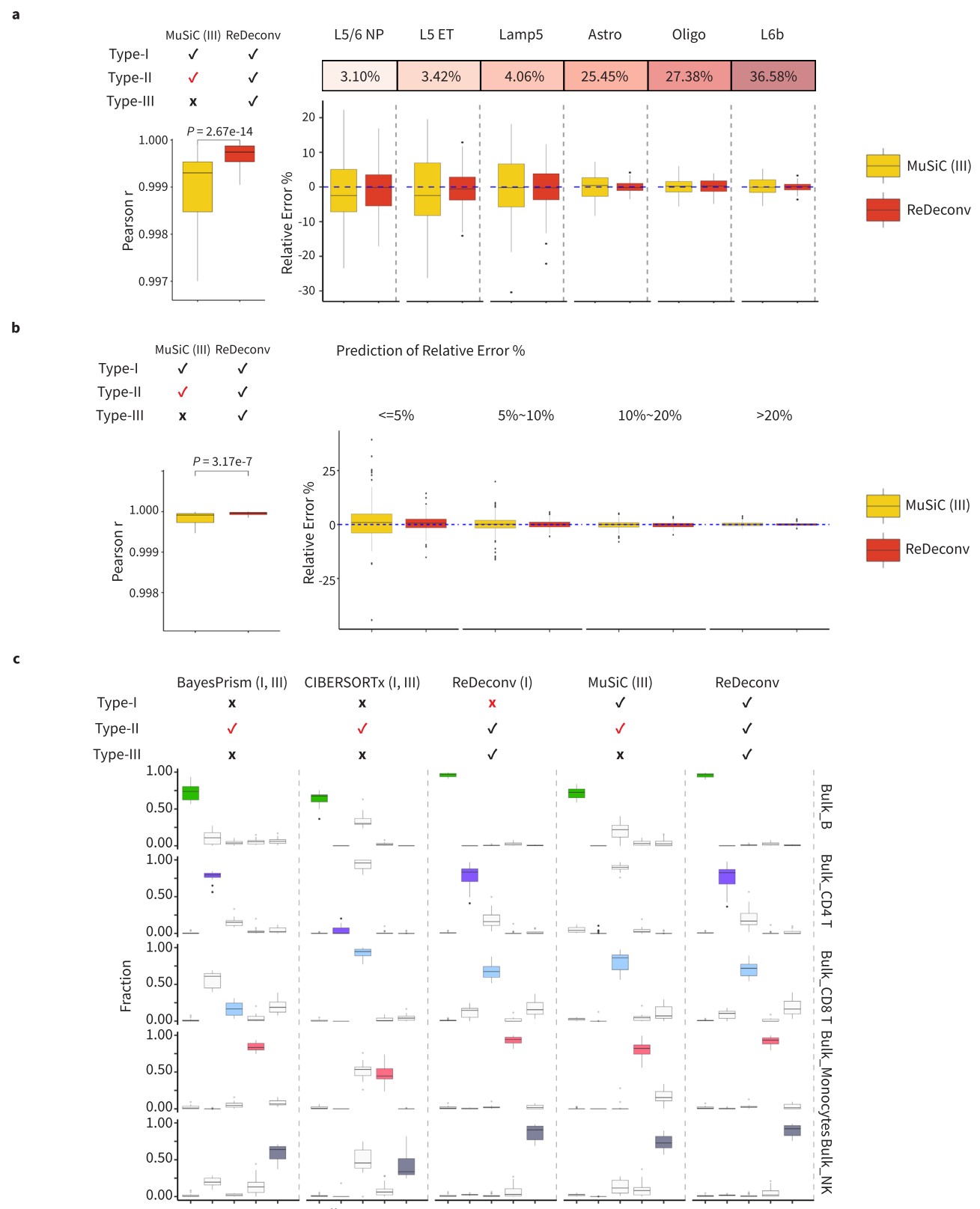

The primary reasons that Type-I and Type-II issues have been unnoticed during the evaluation of current prevalent deconvolution models can be attributed to two factors. First, real tumor data often lacks a definitive ground truth. Second, when synthetic bulk RNA-seq data is derived from either raw-count, CPM, or CP10K scRNA-seq data[18–20,51] that are also utilized as references, resulting in the disappearance of Type-II issues or both Type-I and Type-II issues[18–20,51]. However, it's crucial to note that this type of bulk RNA-seq data usually doesn't exist in real-world scenarios.

In a recent tumor deconvolution benchmarking study[52], researchers created in vitro bulk mixtures by combining RNAs of purified cells in specific ratios and in silico mixtures by linearly

**Fig. 6 | Examining the impacts of Type-III issues on deconvolution. a, b** The Pearson correlations (left) and relative errors (right) of predictions to ground truth by MuSiC (III), a modified version of MuSiC suffering from Type-III issues only, and ReDeconv. The synthetic data was (**a**) 100 synthetic bulk RNA-seq data with the same fractions for each cell type in which the fractions were generated randomly for different cell types (SYN Data E) and (**b**) 100 synthetic bulk RNA-seq data with the random fractions for all cell types and all samples (SYN Data F). **c** Comparison of the results from ReDeconv, ReDeconv (I) (with Type-I issues), BayesPrism[18] (I, III) (with Type-I, III issues), CIBERSORTx (I, III) (with Type-I, III issues), and MuSiC (III) (with Type-III issues) applied to real bulk RNA-seq data of sorted PBMC immune

cells ($n = 17$ for each cell type of sorted bulk samples), including B, CD4T, CD8T, Monocytes and NK cells, from COVID-19 viremia patients (GSE216529). Another bulk RNA-seq data of sorted PBMC cell types from Zika virus-infected patients (GSE13228) was used as the reference. In this sorted bulk RNA-seq data, the impact of Type-I issues on the deconvolution is limited. In the box plots presented in this figure, the values are depicted as the median, represented by the middle line, and the 25th and 75th percentiles, represented by the box. For the comparison of the Pearson Correlation coefficient 'r' between two populations in Fig. 6a and b, the *P*-values were determined using a two-sided Wilcoxon rank-sum test. Source data are provided as a Source Data file.

combining TPM counts of RNA-seq data of sorted cells and used bulk sorted RNA-seq TPM data as the reference for deconvolution. In such a case, Type-I and Type-II issues were unlikely to occur as the same transcriptome size was assumed when the bulk mixture data was generated. It's also worth noting that when the genes used to normalize reference RNA-seq data (including both scRNA-seq and sorted bulk RNA-seq data) and to normalize sorted bulk RNA-seq data used to generate in silico mixture RNA-seq data were different, it could also introduce an expression bias between references and mixtures, which may significantly affect the deconvolution results.

Bulk cellular deconvolution employs a comprehensive strategy that leverages gene expression of cell types from a scRNA-seq reference to align the expression profiles of bulk samples. As a result, the accuracy primarily relies on the degree of gene expression similarity between identical cell types in bulk samples and single-cell reference. Rather than solely seeking signature genes capable of distinguishing different cell types, ReDeconv also assesses their expression stability based on expression variance size. Only those expressions that remain stable within a specific cell type are expected to exhibit minimal differences in their expression between bulk samples and single-cell reference. ReDeconv therefore utilizes these stable signature genes to improve the accuracy of our predictions.

CIBERSORT/CIBERSORTx attempts to find cell type fractions in such a way that the distance between $x_i$ and its predicted value, $\sum_{t=1}^{n} f_t \mu_{it}$, should be small for all signature genes. The training algorithm (Support Vector Regression or SVR) for the CIBERSORT/CIBER-SORTx model ignores some small distances. However, a large distance is always a problem. BayesPrism[18] and MuSiC[20] are likely to encounter similar issues as they also only utilize mean information from scRNA-seq data used as references. In ReDeconv, we try to find the solution with the maximum overall probability given by Eq. (1). Therefore, a large distance between $x_i$ and its prediction value, $\sum_{t=1}^{n} f_t \mu_{it}$, would not be a problem if the probability of having this distance is high. However, a small distance would also be a concern if the probability of having this distance is low, implying that this small distance should be even smaller.

Potential future directions for ReDeconv include the implementation of two improvements. First, it remains unclear whether ReDeconv can effectively remove the "batch effects" and cluster cells of the same type from different samples together by applying the CLTS normalization. It is evident that ReDeconv has a significant advantage in this regard by effectively reducing technology-derived effects. Second, ReDeconv currently does not provide inferred expression profiles while outputting the proportions of cell types. We plan to further optimize and provide this functionality in the next version of ReDeconv.

## Methods
### A method for scRNA-seq data normalization that can address the technology-derived effects and avoid the problems of CPM/CP10K normalization
We mainly utilize the linear relation of transcriptome size means for different cell types from any two samples to perform the scRNA-seq

data normalization (Fig. 2a, Supplementary Fig. 2b, c; Supplementary Fig. 3). Suppose that a scRNA-seq data includes $S$ samples and $n$ cell types; first, for each sample $i$ and cell type $j$, we compute $x_{ij}$, the transcriptome size mean for cells of cell type $j$ in the sample $i$ (if the sample $i$ has cells belonging to cell type $j$). Then, we fix a sample, such as sample 1, as the baseline and compute the linear correlation coefficients $a_i$ and $b_i$ for any sample $i$ to fit the following linear relation:

$$x_{ij} = a_i x_{1j} + b_i, 1 \le j \le n$$

Next, we find $v_{\max}$, the maximum of all $b_i/a_i$, $1 \le i \le S$ (Note : $a_1 = 1$, $b_1 = 0$). Finally, we update the expression $u_{icg}$ for gene $g$ of cell $c$ in sample $i$ with $\frac{u_{icg}}{a_i} + \frac{v_{\max} - b_i/a_i}{G}$, where $G$ is the number of genes in the expression data. By adding $\frac{v_{\max}}{G}$ to the expression of each gene in every cell of sample i, we prevent the emergence of negative expression values after the shift, $-\frac{b_i}{a_i}$, which is uniformly distributed across all genes. After the normalization, transcriptome size means that the same cell type in any two samples will be similar (Fig. 2a, c, Supplementary Fig. 2e).

**Note 1**: When employing Seurat for cell type identification, there can be instances where multiple clusters are inaccurately assigned the same cell type due to the use of inappropriate marker genes. This can result in the merging of loosely related cells into a single cell type, significantly impacting the linear relationship between the mean transcriptome sizes of diverse cell types in various samples. In such cases, we recommend using Seurat clusters as a substitute for cell types when carrying out scRNA-seq data normalization. Subsequently, the normalized data obtained from our innovative method can be used to pinpoint marker genes and establish cell types for clusters.

**Note 2:** In the default setting of ReDeconv, sample-0 (the first in the sorted sample list) is used as baseline sample. If all samples within a scRNA-seq reference dataset are under the same or similar conditions and all cells within the data are properly annotated, a strong linear correlation should exist between the mean cell type transcriptome sizes of any two samples. Under CLTS-normalization, the expression profiles of all cells within the same sample are primarily amplified or suppressed at the same ratio, which minimally impacts the relative mean cell type transcriptome size, regardless of which sample is used as the baseline.

However, there are cases where the linear correlation between the mean cell type transcriptome sizes of some sample pairs in the scRNA-seq reference is weak due to different conditions or batches. To handle such cases, in Step 2 of the ReDeconv program (as detailed in the online tutorial), we generate a file named 'Extra_information.txt'. This file encompasses information such as under a given threshold of transcriptome size correlation (0.95 as default), which samples could be merged. The Pearson correlation between the mean cell type transcriptome sizes of any merged sample and the baseline sample should exceed the threshold. In such cases, we suggest users utilize this file and feature to merge some samples and determine the most suitable baseline sample.

## How the Type-I and Type-II issues impact the cell type deconvolution

The most import hypothesis for the cell type deconvolution is that if a given mixture sample includes $n$ cell types, and for any gene $i$ (for at least $m$ chosen signature genes), **1)** its expression value $x_i$ in the mixture sample is known; and **2)** its expression means $\mu_{i1}, \mu_{i2}, \ldots, \mu_{in}$ in all cell types are also given. Then for the $m$ signature genes, we should have

$$
\begin{aligned}
f_1 u_{11} + f_2 u_{12} + \ldots + f_n u_{1n} &\approx x_1 \\
f_1 u_{21} + f_2 u_{22} + \ldots + f_n u_{2n} &\approx x_2 \\
&\ldots\ldots\ldots\ldots \\
f_1 u_{m1} + f_2 u_{m2} + \ldots + f_n u_{mn} &\approx x_m
\end{aligned}
\tag{1}
$$

where $f_1, f_2, \ldots, f_n$ are fractions or coefficients of these cell types in the mixture sample and each equation is for a gene. Then, deconvolution models attempt to find all coefficients to fit the Eq. (1), where models based on linear regression, such as CIBERSORT/CIBERSORTx[19,25], try to directly fit the Eq. (1), while other models based on Bayesian or probability, such as BayesPrism[18], MuSiC[20], and our model, assume that the expected value of $f_1 \mu_{i1} + f_2 \mu_{i2} + \ldots + f_n \mu_{in}$ should be equal to $x_i$, for $1 \leq i \leq m$. After we obtain the Eq. (1), we can then study how Type-I or Type-II issues impact cell type deconvolution.

$$
\begin{aligned}
(f_1/r_1)(r_1 u_{11}) + (f_2/r_2)(r_2 u_{12}) + \ldots + (f_n/r_n)(r_n u_{1n}) &\approx x_1 \\
(f_1/r_1)(r_1 u_{21}) + (f_2/r_2)(r_2 u_{22}) + \ldots + (f_{1n}/r_n)(r_n u_{2n}) &\approx x_2 \\
&\ldots\ldots\ldots\ldots \\
(f_1/r_1)(r_1 u_{m1}) + (f_2/r_2)(r_2 u_{m2}) + \ldots + (f_{n1}/r_n)(r_n u_{mn}) &\approx x_m
\end{aligned}
\tag{2}
$$

If we have applied CPM/RPM to scRNA-seq data, then the expression means of all genes in different types of cells would be amplified or shrunk, where expression means of genes in cell types with small transcriptomics sizes would be amplified. To maintain the balance of Eq. (1), all coefficients must also be adjusted according to Eq. (2), where $r_1, r_2, \ldots r_n$ are constants to increase or reduce the expression means of all genes in different types of cells.

So, it is easy to understand that if the expression average of a cell type obtained from the scRNA-seq data is amplified $r$ times, then its fraction should be shrunk $r$ times correspondingly. Simulations showed that the predicted percentages of cell types are strongly associated with their cell type transcriptome sizes (Fig. 5a, Supplementary Fig. 11). (**Note:** Although $r_1, r_2, \ldots r_n$ for different genes should not be the same, they have similar values. So, Eq. (2) can be used to estimate the new fractions caused by CPM/CP10K normalization, or Type-I issues.)

$$
\begin{aligned}
f'_1 u_{11} + f'_2 u_{12} + \ldots + f'_n u_{1n} &\approx L_1 x_1 \\
f'_{11} u_{21} + f'_2 u_{22} + \ldots + f'_n u_{2n} &\approx L_2 x_2 \\
&\ldots\ldots\ldots\ldots \\
f'_1 u_{m1} + f'_2 u_{m2} + \ldots + f'_n u_{mn} &\approx L_m x_m
\end{aligned}
\tag{3}
$$

When gene length normalization (RPK) is applied only to the bulk RNA-seq data of the mixture sample, the expression of each gene will be divided by its gene length. This will result in the right side of each equation in (1) being divided by its corresponding gene length. Similarly, if gene length normalization is applied only to the scRNA-seq data, the left side of each equation in (1) will be divided by its corresponding gene length, which is equivalent to multiplying the right side of each equation in (1) by the corresponding gene length and obtaining Eq. (3). This will cause the predicted fractions to change greatly, unlike the case of CPM/CP10K normalization, where we would not know how the predicted fractions would be altered. It is clear that in this case, the fraction predictions are also impacted by the signature genes chosen, i.e. different signature genes will lead to different results.

To prevent the problems caused by gene length normalization, or Type-II issues, we need to ensure that, after normalization, either *both the scRNA-seq data and RNA-seq data are related to the gene length*, or *neither of them are related to the gene length*. If we do not carefully check the technology differences for obtaining scRNA-seq and bulk RNA-seq data, we may have Type-II issues when utilizing some previous methods[18–20,25] for cell type deconvolution. For example, if we let $r_g$ be the RNA count of gene $g$, or the actual number of RNAs expressed for the gene $g$, in a sample/cell, then the measure value for the expression of $g$ in the sample/cell is $cr_g$ when using 10X Genomics Chromium system (for scRNA-seq data) and $c'r_g L_g$ when using Illumina HiSeq 2000 (for total bulk RNA-seq data), where $c$, $c'$ are constants and $L_g$ is the RNA length of the gene $g$. So, if we apply RPK to both the scRNA-seq and bulk total RNA-seq data, then we will have Type-II issues for the deconvolution, where we should only apply RPK to the total bulk RNA-seq data to prevent this type of issues. Therefore, before we decide if we should apply RPK to scRNA-seq or (bulk) RNA-seq data, we need to check the techniques used to obtain the transcriptomics in detail.

From Eq. (1), it is easy to see that if we apply CPM/CP10K to bulk RNA-seq data for mixture samples, we would multiply the expression of each gene by a fixed constant, such as $c = \frac{1,000,000}{\sum_{g=1}^{G} x_g}$ for CPM normalization, or the right side of each equation in (1) would be multiplied by this fixed constant, where G is the total number of gene and $x_g$ is the expression of gene g. Therefore, we do not need to perform normalization to the scRNA-seq data and the percentages of all cell types from computation would not be changed, even though all fractions from computation would be amplified by this fixed constant. Finally, we can conclude that in cell type deconvolution, we should apply RPK, RPKM (CPM + RPK), or TPM (RPK + CPM) to total bulk RNA-seq data. We should use raw count scRNA-seq data if the scRN-seq data is from only one sample; otherwise, we should use our method to normalize the data (refer to Supplementary Table 2).

Utilizing incorrectly formatted scRNA-seq and bulk RNA-seq data, such as when both types of data are in CPM or TPM format, in any deconvolution models may lead to both Type-I and Type-II issues. Under such circumstances, the prediction of expected cell type fractions can be estimated by the following equations, which are a combination of Eqs. (2) and (3).

$$
\begin{aligned}
(f'_1/r_1)(r_1 u_{11}) + (f'_2/r_2)(r_2 u_{12}) + \ldots + (f'_n/r_n)(r_n u_{1n}) &\approx L_1 x_1 \\
(f'_1/r_1)(r_1 u_{21}) + (f'_2/r_2)(r_2 u_{22}) + \ldots + (f'_n/r_n)(r_n u_{2n}) &\approx L_2 x_2 \\
&\ldots\ldots\ldots\ldots \\
(f'_1/r_1)(r_1 u_{m1}) + (f'_2/r_2)(r_2 u_{m2}) + \ldots + (f'_n/r_n)(r_n u_{mn}) &\approx L_m x_m
\end{aligned}
\tag{4}
$$

## A probability model for cell type deconvolution to address Type-III issues

Suppose that in a mixture sample with bulk RNA-seq data, there are $n$ cell types, with $k_1, k_2, \ldots, k_n$ cells in each cell type, respectively. Further suppose that given a fixed gene $i$, the expression $x_{ij}^t$ for any cell $j$ of a fixed cell type $t$ follows the same distribution (not necessarily to be a normal distribution) with $\mu_{it}$ as the mean and $\sigma_{it}^2$ as the variance, which can be obtained from scRNA-seq data.

If the conditions for samples with scRNA-seq data and bulk RNA-seq data are very similar, then we can assume that for any given gene, the expression of the gene in the cells of the same cell type in the reference samples with scRNA-seq data and mixture samples with bulk RNA-seq data follows the same distribution.

As the number of cells for each cell type is usually large in scRNA-seq data (as we can try to find scRNA-seq data with a good size as references), we can assume that the estimated values of mean and variance are equal to those for the actual distributions.

Let $g_i$ be the expression of gene $i$ in the mixture sample, then

$$g_i = \left(x_{i1}^1 + x_{i2}^1 + \ldots + x_{ik_1}^1\right) + \left(x_{i1}^2 + x_{i2}^2 + \ldots + x_{ik_2}^2\right) + \ldots$$
$$+ \left(x_{i1}^n + x_{i2}^n + \ldots + x_{ik_n}^n\right)$$

As for any cell type $t$, usually $k_t$ is large for the bulk RNA-seq data, $x_{i1}^t + x_{i1}^t + \ldots + x_{ik_t}^t$ should closely follow the normal distribution $N\left(k_t\mu_{it}, k_t\sigma_{it}^2\right)$. Then the expression of gene $i$ in the mixture sample, $g_i$, should closely follow the normal distribution $N\left(\sum_{t=1}^n k_t\mu_{it}, \sum_{t=1}^n k_t\sigma_{it}^2\right)$.

Let $x_i$ be the expression (measured) value of gene $i$ in the bulk RNA-seq data, then there is a constant $c$ such that $x_i = cg_i$. So, $x_i$ should closely follow the normal distribution $N\left(c\sum_{t=1}^n k_t\mu_{it}, c^2\sum_{t=1}^n k_t\sigma_{it}^2\right)$.

$$\text{Let } \mu_i = \sum_{t=1}^n k_t\mu_{it}, \ \sigma_i^2 = \sum_{t=1}^n k_t\sigma_{it}^2.$$

$$\text{then, } x_i \sim N\left(c\mu_i, c^2\sigma_i^2\right).$$

Let $f_t = ck_t$, then $c\mu_i = \sum_{t=1}^n f_t\mu_{it}$ and $c^2\sigma_i^2 = c\sum_{t=1}^n f_t\sigma_{it}^2$. So, we have the following density function for $x_i$.

$$p\left(x_i = cg_i | c\mu_i, c^2\sigma_i^2\right) = \frac{1}{\sqrt{2\pi c^2\sigma_i^2}} \exp\left(-\frac{(x_i - c\mu_i)^2}{2c^2\sigma_i^2}\right)$$

We choose $m$ most relevant genes for doing the cell type deconvolution. Then we have the following joint probability.

$$f(f_1, f_2, \ldots, f_n) = \prod_{i=1}^m \left(\frac{1}{\sqrt{2\pi c^2\sigma_i^2}} \exp\left(-\frac{(x_i - c\mu_i)^2}{2c^2\sigma_i^2}\right)\right)$$

$$= \prod_{i=1}^m \left(\frac{1}{\sqrt{2\pi c\sum_{t=1}^n f_t\sigma_{it}^2}} \exp\left(-\frac{\left(x_i - \sum_{t=1}^n f_t\mu_{it}\right)^2}{2c\sum_{t=1}^n f_t\sigma_{it}^2}\right)\right) \quad (5)$$

Finally, find the fractions $f_1, f_2, \ldots, f_n$ of different cell types in the mixture sample such that $f(f_1, f_2, \ldots, f_n)$ is maximized.

## Choose stable signature genes

There are two important considerations when choosing signature genes. First, there should be genes for each cell type that can be used to distinguish the given cell type from all other cell types. Second, the expression levels of these genes should maintain stability within each cell type. As a result, the expression of these genes from the same cell type in both reference and mixed samples should exhibit a high degree of similarity, which is helpful in reducing Type-III issues.

We have two steps to identify signature genes. First, we choose gene candidates, called initial signature genes, such that for each cell type, we find some genes that can be used to separate this cell type from others. For any gene, we group the expressions of the gene by cell types. Then, we use a t-test and mean fold changes to check the expressions of genes in all pairs of groups. If, for any given cell type, 1) the p-value of the t-test between the expressions of the gene in the group for the given cell type and each of all other cell types is less than 0.05, and 2) the mean fold change of gene, relative to any other cell types, should surpass the threshold (while the default value is set at 2, it can be adjusted based on your specific data), then we say that this gene is a level-0 signature gene for the given cell type.

If only one of the other cell types cannot pass the t-test and fold-change threshold, we say that this gene is a level-1 signature gene for this given cell type. Similarly, if only two of the other cell types cannot pass the t-test and fold-change threshold, we say that this gene is a level-2 signature gene for the cell type. We found that in most cases, the number of level-0 signature genes is big enough for choosing reliable signature genes in the next step. As this step needs to process a large number of cells and genes, we record more candidates in case there are not enough level-0 initial signature genes for some cell types. You can also use the method of CIBERSORTx, expressions in one group versus expressions in all other groups as a whole, to choose initial signature genes, which can reduce the running time.

Then, we further choose genes from the initial signature genes (from level-0 first) for each cell type such that their expressions are stable in the given cell type. To do this, we compute the ratio of the standard deviation to the mean of expressions in the given cell type and choose the top $k$ genes with the smallest ratios. The following is the reason why we choose genes with the smallest ratios. Suppose gene $i$ is an initial signature gene for cell type $t$, then the expressions of all cells for the gene in the cell type $t$ should closely follow a normal distribution:

$$\left(x_{i1}^t + x_{i2}^t + \ldots + x_{ik_t}^t\right) \sim N\left(k_t\mu_{it}, k_t\sigma_{it}^2\right)$$

If we let $f_t\mu_{it}$ be equal to $x_{i1}^t + x_{i2}^t + \ldots + x_{ik_1}^t$ then $\frac{f_t}{k_t} \sim N\left(1, \frac{\sigma_{it}^2}{\mu_{it}^2 k_t}\right)$, where $f_t$ is the estimated fraction or cell number of the cell type $t$ and $k_t$ is the true number of cells in cell type $t$. Therefore, we know that a small ratio $\frac{\delta_{it}}{\mu_{it}}$ means that $\frac{f_t}{k_t}$ has a bigger probability of being within a small distance to 1; that is, we should choose signature genes with a small ratio $\frac{\delta_{it}}{\mu_{it}}$ for cell type $t$. We found that, usually, using this way, we only need to choose about 100 signature genes for each cell type, which is good enough to guarantee the performance of the deconvolution.

## Generate synthetic bulk RNA-seq data

Each synthetic bulk RNA-seq data was generated from a *raw count* scRNA-seq data such that all cells are from only *one sample*. After setting the number of cells for all cell types, we randomly selected the given number of cells for all cell types and merged their expression profiles by adding the expression of each gene in all chosen cells together, which obtains the synthetic bulk RNA-seq data in RPK. Then, we applied CPM normalization to the RPK synthetic bulk RNA-seq data to obtain TPM of the data. We obtained the raw count synthetic bulk RNA-seq data by multiplying expression of each gene in the RPK synthetic RNA-seq data with its gene length. We repeated this process 100 times to generate 100 mixture samples for each synthetic bulk RNA-seq data set (Supplementary Data 1, 2).

## Note

Do not use CPM or CP10K scRNA-seq data to generate synthetic bulk RNA-seq data as in the real world, transcriptome sizes of cells usually are not equal. If you use expression profiles of cells from two or more samples to generate synthetic bulk RNA-seq data, you may have the problem that transcriptome sizes of the same types of cells in the data change too much, such as changing multiple folds, which is usually not true in the real world. If cells of the same type in both the reference and mixed samples originate from the same cell population, then the mean expression of each gene in the same cell type is likely to be identical in the reference and mixture samples. Therefore, if the scRNA-seq data used to generate synthetic bulk RNA-seq data is also used as a reference for deconvolution, the impact of Type-III issues on the deconvolution would be minimal.

## Evaluation of normalization

We selected scRNA-seq data from two samples that were significantly influenced by technology-derived effects. This was noticeable in the transcriptome size of identical cell types from two distinct samples, exhibiting a significant disparity. We also focused on two cell types with considerable differences in their transcriptome sizes. Furthermore, a significant number of cells from these two cell types should be present in both the scRNA-seq and CoxMx data. Our objective was to assess how various normalization techniques affect gene expression and the differentially expressed genes in these two cell types. For the comparison of performance, we utilized the CosMx data as the benchmark or 'ground truth'.

**Compare the expressions of genes under raw count, CP10K-normalized and CLTS-normalized scRNA-seq data.** We examined gene expressions in two types of cells from two samples, considering raw-count, CP10K-normalized, and CLTS-normalized scRNA-seq data. Our aim was to determine if these normalization methods could effectively mitigate technology-derived effects and have the problem of amplifying gene expression in diverse cell types with unequal ratios. (Fig. 2d–f; Supplementary Fig. 5)

**Assess the fold-change directions of genes in two cell types under both CP10K and CLTS normalizations.** This involved determining whether the genes were up-regulated or down-regulated when comparing one cell type to another under the scRNA-seq data normalized via CP10K and CLTS, respectively. Subsequently, we compared these results with the CosMx data to ascertain if the outcomes from CP10K or CLTS normalizations were in closer agreement with the CosMx data. (Fig. 3a, b; Supplementary Fig. 7b, c)

**Conduct an evaluation of significantly differentially expressed genes (DEGs) identified solely under one normalization method.** Using a fold-change threshold of 1.5 and a p-value threshold of 0.05, as determined by the Wilcoxon rank-sum test, we identified significant DEGs in two cell types under CP10K-normalized, but not under CLTS-normalized, scRNA-seq data. We then cross-verified these DEGs with the CosMx data (Fig. 3c-h). We replicated this process for significant DEGs detected under the CLTS-normalized, but not under the CP10K-normalized scRNA-seq data. (Supplementary Fig. 7d–i)

## Evaluations of deconvolution

Our method for deconvolution was evaluated with synthetic and real bulk RNA-seq data, where we compared the performance of ReDeconv with those of BayesPrism, CIBERSORTx, and MuSiC. We have compared the performance of different methods under their defaulting settings. We also tested how Type-I, Type-II, and Type-III issues impact deconvolution, respectively, where different types of issues are introduced by using different combinations of references and mixtures as the Supplementary Table 3.

**Evaluate BayesPrism, CIBERSORTx, MuSiC, and ReDeconv for their recommended input data format and default settings.** We used CLTS-normalized (raw count if all cells are from on sample) scRNA-seq and TPM bulk RNA-seq data as inputs for ReDeconv, and raw count scRNA-seq and bulk RNA-seq as inputs for other three models. Under default settings, BayesPrism and CIBERSORTx have Type-I, Type-II, and Type-III issues, MuSiC has Type-II and Type-III issues, ReDeconv does not have these three types of issues. (Fig. 4; Supplementary Fig. 9s,10)

**Evaluate the impact of Type-I issues to deconvolution.** We used raw count scRNA-seq data and TPM bulk RNA-seq data for BayesPrism and CIBERSORTx, CPM scRNA-seq data and TPM bulk RNA-seq data for MuSiC and ReDeconv. In this case, ReDeconv has Type-I issues, the other three models have Type-I and Type-III issues. However,

when a model has both Type-I and Type-III issues, if the impact of Type-III issues is small, then the Type-I issues dominate the results. So, in this test, we can see how Type-I issues impact the deconvolution – *amplifying fractions of cell types with bigger transcriptome sizes and shrinking fractions of cell types with smaller transcriptome sizes.* (Fig. 5a; Supplementary Fig. 11; ReDeconv (I) in Fig. 5b, c and Supplementary Fig. 12)

**Evaluate the impact of Type-II issues to deconvolution.** BayesPrism and CIBERSORTx were not evaluated in this case as they always have Type-I issues. We used raw count scRNA-seq and bulk RNA-seq data as inputs for MuSiC and CLTS-normalized scRNA-seq and raw count bulk RNA-seq data as inputs for ReDeconv, where ReDeconv has Type-II issues, MuSiC has Type-II and Type-III issues. When MuSiC has Type-II and Type-III issues, Type-II issues should have a dominant impact to the deconvolution when expression of the same type of cells in reference and synthetic bulk have the same distribution. (MuSiC in Fig. 4 and Supplementary Figs. 9, 10a, b; ReDeconv(II) in Fig. 5b, c and Supplementary Fig. 12)

**Use ReDeconv to evaluate the impact of Type-I and/or Type-II issues on deconvolution.** As BayesPrism, CIBERSORTx and MuSiC do not address Type-III issues, we used ReDeconv to test how Type-I and/or Type-II issues impact deconvolution by using different combinations of scRNA-seq and bulk RNA-seq data as inputs, where using CPM scRNA-seq and TPM bulk RNA-seq data would introduce Type-I issues, raw count scRNA-seq and bulk RNA data as inputs would introduce Type-II issues, CPM scRNA-seq and raw count bulk RNA-seq data as inputs would introduce both Type-I and Type-II issues. (Fig. 5b, c; Supplementary Fig. 12)

**Evaluate the impact of Type-III issues on deconvolution.** We first tested ReDeconv and MuSiC on synthetic and real bulk RNA-seq data with diverse fractions for different types of cells, which can also check if cell type fractions in the mixture samples affect the prediction accuracies. We used raw count scRNA-seq and TPM bulk RNA-seq data as input. So, both models do not have Type-I and Type-II issues. The major difference is that the MuSiC model has Type-III issues and ReDeconv does not have Type-III issues. (Fig. 6a, b; Supplementary Figs. 14, 15, ReDeconv in Fig. 4d)

We then tested BayesPrism, CIBERSORTx, MuSiC, and ReDeconv on real bulk sorted RNA-seq data, where we chose input combinations that have no Type-II issues for all methods, such as using raw count scRNA-seq and TPM bulk sorted RNA-seq as inputs. Although BayesPrism and CIBERSORTx always have Type-I issues, in the bulk sorted RNA-seq data, Type-I issues have limited impact on the deconvolution for the dominant cell types (we specially tested this by introducing Type-I issues to Redeconv). So, in the tests, the major difference is that BayesPrism, CIBERSORTx, and MuSiC have Type-III issues while ReDeconv addresses this type of issues. For a fixed bulk sorted RNA-seq data, we tested four models using the same bulk sorted RNA-seq, another bulk sorted RNA-seq, and scRNA-seq data as references, respectively. These tests can check the performance of all four models in terms of stability. (Fig. 6c; Supplementary Fig. 16)

## Note

In the case of using bulk sorted RNA-seq data as references, the TPM format is equivalent to CPM scRNA-seq data. We obtained the data format that is equivalent to the raw count scRNA-seq data by adjusting TPM RNA-seq data with transcriptome sizes of corresponding types of cells.

## Reporting summary

Further information on research design is available in the Nature Portfolio Reporting Summary linked to this article.

## Data availability

All scRNA-seq, synthetic and real bulk RNA-seq data used for evaluations are available at https://redeconv.stjude.org. All other relevant data supporting the key findings of this study are available within the article and its Supplementary Information files. CosMx for mouse brain is available at the Zenodo data repository under record number 10520022. Source data are provided with this paper.

## Code availability

The source codes for ReDeconv are freely available online at GitHub: https://github.com/jyyulab/redeconv and zenodo repository https://doi.org/10.5281/zenodo.14589212[53]. The documentation with a tutorial is available online at https://redeconv.stjude.org. The demo codes about how to integrate ReDeconv with Seurat, Scanpy, and scMINER for scRNA-seq data cell type annotations and other analyses are available at https://redeconv.stjude.org/software.

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

## Acknowledgements

We thank the members of the Yu Lab for testing and improving ReDeconv, Anuj Jain for helping with the portal development, and Sarah August for scientific editing. This work was supported in part by National Institutes of Health grants R01GM134382, R01CA274251, U01CA264610, U01CA281868, and RF1AG068581, and by the American Lebanese Syrian Associated Charities. The content is solely the responsibility of the authors and does not necessarily represent the official views of the National Institutes of Health.

## Author contributions

S.L. and J.Yu conceived of the project and wrote the manuscript. S.L. developed and implemented ReDeconv and performed data analyses and model evaluations. J.Yang assisted model evaluation and wrote the manuscript. L.Y. created the web portal with help from R.J. J.L. and J.W. assisted with model evaluation. All authors commented on the manuscript at all stages.

## Competing interests

The authors declare no competing interests.
