## [Transparent Peer Review file · Nature Communications]

Transcriptome size matters for single-cell RNA-seq normalization and bulk deconvolution

Corresponding Author: Dr Jiyang Yu

Version 0:

Reviewer comments:

Reviewer #1

(Remarks to the Author)

The manuscript by Lu et al. aims to show the advantages of their newly developed method ReDeconv, which uses a new normalization method (CLTS) to correct differences in transcriptome size into scRNA-seq normalization and deconvolution. Along with the results shown in the manuscript, the authors show the outperformance of their method as compared to other well-known deconvolution software. However, some points may need clarification/improvements from the authors:

- Validation of Transcriptome Size Differences: In Figure 1, the authors attempt to demonstrate that differences in transcriptome size are cell type-specific rather than technical artifacts. They base this on the assumption that "technology-derived effects would affect all cells within a sample equally." However, this assumption may not hold true. Different cell types can express distinct sets of transcripts, and sequencing protocols can introduce biases based on transcript sequence composition and other factors. Therefore, different cell types might indeed be affected differently by technical biases. To substantiate their claim, the authors should compare data across various protocols. The correlation of the two samples shown in Figures 1a-c is insufficient to conclusively state that "technology-derived effects proportionally amplify the transcriptome sizes of all cells in the same sample," as this can be protocol-dependent. Additional comparisons are necessary to strengthen this claim.

Furthermore, if more than two samples are included in an analysis, it is unclear which sample is used as the reference. Is it random? How discrepancies between different comparisons are addressed?

- Benchmarking with Tumor Deconvolution DREAM Challenge Data: For an unbiased evaluation of deconvolution software, the Tumor Deconvolution DREAM Challenge consortium has created several datasets with various in vitro and in silico mixtures. These datasets should be included in the benchmarking of the authors' approach (<https://www.biorxiv.org/content/10.1101/2022.06.03.494221v2>). The current in silico benchmark in Figures 4a-c does not follow the same trend as the results for real bulk-RNAseq data in Figures 4d-e, indicating a potential discrepancy that needs to be addressed.

- In Figure 4a, ReDeconv shows a perfect prediction of the ground truth fractions with minimal deviation. However, in Figure 5b (first column), where deconvolution of the same dataset is performed, a relative error of around 10% is reported. This inconsistency needs to be explained.

- The authors should benchmark against BLADE deconvolution software (<https://www.nature.com/articles/s41467-021-26328-2>), which also addresses the issue of variance size (Type III).

Overall, while ReDeconv shows potential, the manuscript would benefit from addressing these points to provide a more robust and convincing validation of the method.

(Remarks on code availability)

I did not review the code per se, but I did run the website of the software using the test data. The website worked properly, and I got the expected results, within a reasonable frame of time.

In principle, it seems a suitable resource for the community.

Reviewer #2

(Remarks to the Author)

In this manuscript, the authors developed a novel algorithm, ReDeconv, to normalize scRNA-seq data and deconvolute bulk RNA-seq. ReDeconv corrects differentially expressed genes typically misidentified by CP10K method and enhances the accuracy of bulk deconvolution by mitigating gene length effect and modeling the expression variation. The authors performed relatively comprehensive evaluation on bulk RNA deconvolution by comparing with the existing tools and demonstrated its advantages. The algorithm has potential interest and could benefit the community. However, the current version of manuscript has the following drawbacks: 1) it lacks a deeper and comprehensive evaluation of scRNA-seq normalization; 2) it exists several unproved or not well explained key hypothesis; and 3) it is illogical and incoherent in some parts of writing. Below are the detailed comments.

My major concerns:

1. It seems that CLTS method in ReDeconv is too much simplified in the potential effect from technical noise. The paper 'Droplet barcoding for single-cell transcriptomics applied to embryonic stem cells' demonstrated technical noises were contributed to the scRNA variability. But from Lines 47-48, the authors claimed 'The observed differences in transcriptome size of scRNA-seq data across a broad range of tissues and species suggest that they are inherent to the nature of biology, rather than being artifacts induced by technology-derived effects'. So, this needs more clarification and prove.
2. Follow point 1, Line 112 'cells of the same type demonstrate considerable similarity in (true) transcriptome sizes.' is not accurate as Extend Data Fig.1 clearly show L5 PT CTX have broad range of transcriptome size, which might be resulted from technical noise. While one of the purposes for using CP10K is to mitigate the technical noise, how CLTS mitigates technical noise is not clear. It looks like be omitted in the current version by making a strong and unproved hypothesis.
3. Based on the description of the method, CLTS is a 'supervised' normalization method since it needs cell type information. In this case, I wondered if CLTS could work on data that cell type is not clearly identified? If not, then the cell types identified by conventional CP10K already introduce a bias (claimed by paper), how does CLTS re-correct the potential misidentification of cell type by CP10K?
4. Based on the description of the method, CLTS used sample 1 (x1j in Supple Methods) as the base reference. In this case, will the normalized transcriptome size change according to different selections of reference sample? If so, how could CLTS determine which sample is the better representer of 'true' transcriptome size?
5. Follow point 3, the authors should provide some rationale why they choose these two samples as an example. More samples are needed in order to demonstrate the advantages of CLTS. And Extend Data Fig. 1b,c needs more explanations and clearer annotation.
6. Line 65, the author mentioned SCnorm and SCTransform also susceptible to the same scaling effect like CP10k. It is better to make a comprehensive comparison by including them in main figs or supple figs.
7. Lines 98-100, 'Consistent with previous reports, we observed that cells of a single type in the same sample exhibit roughly the same transcriptome size, while the transcriptome size varies significantly across different cell types'. It is not clear the meaning of 'a single type in the same sample exhibit roughly the same transcriptome size'.
8. It looks like Fig. 1b missing many types. Could authors further clarify this?
The definition of Type I, II, III issues is not clearly defined in the current version. Are these biologically related issues or technically related issues? For example, how type I issue reflects false-positive in deconvolution scenario?
9. Line 221, Supplementary Fig.2c. Could author provide a real world example to prove?
10. Line 227, 'the expression of each gene in each cell of any given cell type is a random event' needs to be referred or prove.
11. It would be better to move the workflow of the algorithm at the beginning.

My minor comments:

1. Line 45, references are needed for multiple scRNA-seq datasets.
2. More detailed descriptions for Supple Fig. 1 are needed. Current explanation is hard to follow
3. It would be better to only have one terminology system for supplementary materials (extend data Fig in line 106, extended Fig, line 98; supplementary fig Line 107).
4. Same r value in both Fig. 1a and 1c. Is it a typo?
5. More clear explanations are needed for $u_{icg/a_i} + (v_{max-b_i/a_i})/G$.
6. Fig. 2h needs more explanations.

(Remarks on code availability)

Reviewer #3

(Remarks to the Author)

I co-reviewed this manuscript with one of the reviewers who provided the listed reports. This is part of the Nature Communications initiative to facilitate training in peer review and to provide appropriate recognition for Early Career Researchers who co-review manuscripts

(Remarks on code availability)

Reviewer #4

(Remarks to the Author)

(Remarks on code availability)

Version 1:

Reviewer comments:

Reviewer #1

(Remarks to the Author)

We appreciate the authors' efforts to implement the suggested improvements and clarify the manuscript. The changes have significantly benefited the overall quality; however, we believe some points still require further clarification:

Regarding my first comment (1a in author's response) about comparing different protocols, we do see the authors' point that the New Fig. 2b, shows a correlation between protocols, although they are versions of the same protocol.

The correlation of the different protocols shown in the New Supp. Fig 3, partly address the question about other protocols, but we think to correctly do it the authors should show a plot similar to New Fig. 2b with the 9 different cell types sequenced with the different protocols (with the exception of the inDrops one), for both PBMC1 and 2.

Regarding comment 1b, which discusses the choice of reference sample, we agree with the authors that the correlation remains consistent regardless of the selected reference. However, we disagree in that this proves that it has no impact at all. For instance, in Response Fig 1c and d (Sample III and Sample IV as reference respectively), in the left plots (comparison of Sample I and II), the pink sample (I think corresponding to Set, although it's difficult to differentiate between all pink samples in the legend) is the third with the highest transcriptome size in d (see red arrow in the attached image), while in c the same sample is the 7th (or more) sample in transcriptome size (blue arrow in the attached document). If the statement of sample size not being affected by technical artefacts, and only by cell type holds true, this couldn't be the case, so either that statement needs to be revised or this strategy is affecting the processing of the data. Authors need to clarify this matter, as their normalization method is based on that statement.

As for the other points we raised, we appreciate the authors' efforts to address them, and their responses are satisfactory.

(Remarks on code availability)

Reviewer #2

(Remarks to the Author)

The author addressed most of my concerns and clarified some ambiguities. However, a few points are still needed to be clarify.

1. Point #3 remains unclear. I am still uncertain about how so many genes are misidentified by CP10K, yet this does not impact cell type annotation. Could the authors provide a profile of the marker genes identified using CP10K and corrected by CTLS? How many of these marker genes are affected? If most marker genes are impacted, it raises questions about the reliability of cell type annotation for CTLS correction. Conversely, if only a few are affected, how does CTLS enhance downstream analyses such as trajectory analysis, GO, GSEA, annotation, cell-cell interaction analysis, and metacell analysis, as shown in Figure 1a?

2. Figure 2 demonstrates that while transcriptome sizes vary across different cell types, their relative positions along the linear line are similar. If we first quantile the cell sizes and then normalize based on these quantiles, could this approach reduce the dependence of CTLS on cell type annotation. Then CTLS could perform as an alternative to CP10K and mitigate issues that arise from cell type annotation errors associated with CP10K?

3. Github page is not available.

4. The CLI and API documentation should be provided more in details. Additionally, please provide a comprehensive tutorial on integrating ReDeconv with Seurat and Scanpy which would facilitate the usage of the tool in the community.

(Remarks on code availability)

Reviewer #3

(Remarks to the Author)

(Remarks on code availability)

Reviewer #4

(Remarks to the Author)

(Remarks on code availability)

Version 2:

Reviewer comments:

Reviewer #1

(Remarks to the Author)

I appreciate the effort made by the authors to respond to all my concerns, which have been satisfactorily addressed.

(Remarks on code availability)

Reviewer #2

(Remarks to the Author)

The authors have addressed my concerns and comments.

(Remarks on code availability)

Reviewer #3

(Remarks to the Author)

(Remarks on code availability)

Reviewer #4

(Remarks to the Author)

(Remarks on code availability)

Point-by-point response to reviewers' comments

Reviewer #1:

The manuscript by Lu et al. aims to show the advantages of their newly developed method ReDeconv, which uses a new normalization method (CLTS) to correct differences in transcriptome size into scRNA-seq normalization and deconvolution. Along with the results shown in the manuscript, the authors show the outperformance of their method as compared to other well-known deconvolution software. However, some points may need clarification/improvements from the authors:

1a.- Validation of Transcriptome Size Differences: In Figure 1, the authors attempt to demonstrate that differences in transcriptome size are cell type-specific rather than technical artifacts. They base this on the assumption that "technology-derived effects would affect all cells within a sample equally." However, this assumption may not hold true. Different cell types can express distinct sets of transcripts, and sequencing protocols can introduce biases based on transcript sequence composition and other factors. Therefore, different cell types might indeed be affected differently by technical biases.

To substantiate their claim, the authors should compare data across various protocols. The correlation of the two samples shown in Figures 1a-c is insufficient to conclusively state that "technology-derived effects proportionally amplify the transcriptome sizes of all cells in the same sample," as this can be protocol-dependent. Additional comparisons are necessary to strengthen this claim.

Response: Yes, we agree that "different cell types might indeed be affected differently by technical biases" and appreciate the suggestion to compare data across various scRNA-seq protocols. We therefore checked the correlation of cell type transcriptome size from scRNA-seq data using different technical protocols (e.g., 10x Chromium v2, v3, CEL-Seq2, Drop-seq, Seq-Well, InDrops, etc). Overall, we can still observe a strong correlation of cell type transcriptome size among different scRNA-seq protocols, suggesting that the impact of technical biases on

transcriptome size might be limited, especially when the scRNA-seq protocols are reliable.

- 1) In Fig. 2b (previously Fig. 1b), the scRNA-seq data for human brain and mouse brain were not generated using the same protocol: for the human brain samples, “10x Chromium Single Cell 3’ Reagent Kit v3” was used; for the mouse brain, “10x Chromium Single Cell 3’ Reagent Kit v2” was employed.
- 2) In addition to the mouse brain example, we also conducted additional tests using two scRNA-seq datasets of human PBMC samples from a benchmarking study, “Systematic comparison of single-cell and single-nucleus RNA-sequencing methods” (*Nat Biotechnol* 38, 737–746, 2020). In these two datasets, sample "pbmc1" and “pbmc2” underwent scRNA-seq profiling using various methods. The correlation analysis results showed that the transcriptome sizes of 9 human PBMC cell types derived from different scRNA-seq methods, including Chromium v2, v3, Drop-seq, Seq-Well, and inDrops, demonstrated a strong linear correlation with Pearson correlation coefficients significantly exceeding 0.7 with the exception of Drop-seq vs. inDrops in pbmc1 (new Supplementary Fig. 3). The low correlation of Drop-seq vs. inDrops in pbmc1 (new Supplementary Fig. 3a) was likely due to technical issues as the correlation is much stronger similar to other pairs in pbmc2 (new Supplementary Fig. 3b).

We have added the above results into the supplementary figures (new Supplementary Fig. 3) and main text (line 145-150) of the revised manuscript.

1b. Furthermore, if more than two samples are included in an analysis, it is unclear which sample is used as the reference. Is it random? How discrepancies between different comparisons are addressed?

Response: We apologize for the confusion. In default setting of ReDeconv, sample-0 (the first in the sorted sample list) is used as baseline sample. It's a random selection as the baseline sample

Response Fig. 1: The effects of baseline sample selection on transcriptome size in CTLS normalization of scRNA-seq data. The pairwise correlation plots of mean transcriptome size of mouse brain cell types in CTLS-normalized scRNA-seq data using Sample I (a), Sample II (b), Sample III (c) and Sample IV (d) as baseline.

won't affect the relative transcriptome size ratio, as illustrated in Response Fig. 1. If all samples within a scRNA-seq reference dataset are under the same or similar conditions and all cells within the data are properly annotated, a strong linear correlation should exist between the mean cell type transcriptome sizes of any two samples. Under CLTS-normalization, the expression profiles of all cells within the same sample are primarily amplified or suppressed at the same ratio, which minimally impacts the relative mean cell type transcriptome size, regardless of which sample is used as the baseline.

However, there are cases where the linear correlation between the mean cell type transcriptome sizes of some sample pairs in the scRNA-seq reference is weak due to different conditions or batches. To handle such cases, in Step 2 of the ReDeconv program (as detailed in the online tutorial), we generate a file named 'Extra_information.txt'. This file encompasses information such as under a given threshold of transcriptome size correlation (0.95 as default), which samples could be merged. The Pearson correlation between the mean cell type transcriptome sizes of any merged sample and the baseline sample should exceed the threshold. In such cases, we suggest users utilize this file and feature to merge some samples and determine the most suitable baseline sample.

To improve the clarity of baseline sample selection in CLTS process, we have added a “Note 2” in the Methods section – “A new method for scRNA-seq data normalization that can address the technology-derived effects and avoid the problems of CPM/CP10K normalization” in Supplementary Information of the revised manuscript.

2a. Benchmarking with Tumor Deconvolution DREAM Challenge Data: For an unbiased evaluation of deconvolution software, the Tumor Deconvolution DREAM Challenge consortium has created several datasets with various *in vitro* and *in silico* mixtures. These datasets should be included in the benchmarking of the authors' approach (<https://www.biorxiv.org/content/10.1101/2022.06.03.494221v2>).

Response: Thank you for your suggestion. While we were aware of this Tumor Deconvolution DREAM Challenge, we didn't use it in our original study primarily due to the limited accessibility of the data. At present, only the RNA-seq data for the *in vitro* mixtures can be accessed publicly (GSE199324). The RNA-seq data for the *in silico* mixtures, along with the ground truth information for both *in vitro* and *in silico* mixtures, remains unavailable to the public. However, to overcome this limitation, we adopted the DREAM Challenge's strategy of creating *in silico* mixtures using TPM bulk sorted RNA-seq data and performed benchmarking analysis. To elaborate, we initiated our process by downloading the bulk sorted RNA-seq data used by the Dream Challenge. Subsequently, for each value of k in 1, 10, 20, 30, and 40, we generated synthetic bulk RNA-seq data for 120 samples. The process for each sample was threefold.

- 1) We randomly generated fractions for all cell types.
- 2) The mean expression profile of each cell type was derived from k randomly selected samples of the respective cell type.
- 3) Each *in silico* mixture sample was formulated as a linear combination of fractions and mean expression profiles of different cell types.

Finally, we applied each of the four methods to the newly generated *in silico* mixtures using the same bulk sorted RNA-seq data as a reference and used the correlation with ground truth to evaluate the performance. Consistent with our other benchmarking analysis, ReDeconv outperformed the other methods significantly in all cases (Response Fig. 2). Of note, we found

that the performance of all four methods increased as the difference in expression between the references and mixtures decreased (as *k* increased).

It should be noted that the *in vitro* bulk mixture by combining RNAs of purified cells in specific ratios and *in silico* mixtures by linearly combining TPM counts of RNA-seq data of sorted cells assumed the same transcriptome size. In such case, Type-I and Type-II issues were unlikely to occur. Therefore, a significant benefit of this specific type of *in silico* mixture used in this DREAM challenge is the ability to assess solely the impact of Type-III issues on BayesPrism and CIBERSORTx. Because these two models automatically apply CP10K/CPM normalization to the scRNA-seq or reference data, such an evaluation is only feasible for finding the fractions of diverse types of cells when the mixture samples are highly purified, hence, reducing the influence of Type-I issues on the deconvolution.

It's also worth noting that when the genes used to normalize reference RNA-seq data (including both scRNA-seq and sorted bulk RNA-seq data) and to normalize sorted bulk RNA-seq data used to generate *in silico* mixture RNA-seq data were different, it could also introduce an expression bias between references and mixtures, which may significantly affect the deconvolution results. Therefore, we have additionally assessed the influence of bias between references and mixtures on deconvolution. In this particular scenario, the references and the bulk sorted RNA-seq data utilized to generate the *in silico* mixtures did not use an same set of genes for normalization.

In this evaluation, we utilized four data sets - R3 (scRNA-seq), R5 (scRNA-seq), R7 (bulk sorted RNA-seq), and R8 (bulk sorted RNA-seq) - to produce and examine this specific kind of *in silico* mixtures. Initially, we identified the genes shared across all four datasets and used them to create four submatrices: R3_sub (CP10K), R5_sub (CP10K), R7_sub (TPM), and R8_sub (TPM). Following this, we used R7_sub to generate 120 *in silico* mixtures, referred to as **Mix_R7_sub**.

Response Fig. 3 | Examination of how expression bias between reference and mixture affects deconvolution. The genes utilized for normalizing the reference samples (both scRNA-seq and bulk sort RNA-seq data) and *in silico* mixture samples were not the same. This discrepancy introduced an expression bias between the reference and mixtures.

Concurrently, we utilized TPM R7 to create an additional set of 120 *in silico* mixtures, termed as **Mix_R7**. In both cases, every corresponding mixture in Mix_R7_sub and Mix_R7 was constructed with the same fractions and bulk samples from R7. Notably, since TPM R7 and TPM R7_sub were not normalized with the same set of genes, an artificial expression bias was introduced between Mix_R7 and the references R3_sub, R5_sub, R7_sub, and R8_sub. In the end, we used R3_sub, R5_sub, R7_sub, and R8_sub as references to determine cell type fractions of Mix_R7 and Mix_R7_sub, respectively. Based on the results, we deduced the following (Response Fig. 3):

- 1) When the expression profiles of each cell type in the reference (R7_sub) and mixtures (Mix_R7_sub) exhibited minimal differences (no artificial expression bias was introduced), all four models had optimal performance.
- 2) The introduced expression bias between the references and mixtures significantly impacted the deconvolution of all methods. This effect was particularly pronounced in cases when the prediction is close to the ground truth (Pearson Correlation coefficient > 0.7).

In summary, ReDeconv still outperformed other deconvolution methods using the evaluation strategy in the Tumor Deconvolution DREAM Challenge. Of note, the use of RNA-seq data from *in silico* and *in vitro* mixtures may bear a few limitations. (1) The ground truth using fractions of RNAs may not reflect the real ground truth of cell fractions of various cell types. (2) Type-I issues will occur when using TPM bulk sorted RNA-seq data or CP10K scRNA-seq data as a reference for real bulk RNA-seq data deconvolution. (3) The potential expression bias between the references and mixtures will be manually introduced when there are differences of features used for the normalization of reference data and used for bulk sorted RNA-seq data.

We appreciate your suggestion of using the DREAM Challenge data for benchmarking of ReDeconv. Accordingly, we have added a paragraph related to the above points in the Discussion section of our revised manuscript (line 448-456).

2b. The current *in silico* benchmark in Figures 4a-c does not follow the same trend as the results for real bulk-RNAseq data in Figures 4d-e, indicating a potential discrepancy that needs to be addressed.

Response: Thank you for your insightful observations and comments. It's plausible that the *in silico* benchmarking results displayed in Figures 4a-c does not follow the same trend observed in the results for actual bulk RNA-seq data presented in Figures 4d-e for the following reasons.

In the synthetic bulk RNA-seq data derived from scRNA-seq data (Figures 4a-c), both the reference and mixture samples consisted of cells from the same population. Thus, the mean expression and standard deviation for each gene and each cell type in the reference and bulk are anticipated to be nearly identical. This suggested that Type-III issues have a minimal effect on the results depicted in Figures 4a-c. This conclusion is further supported by the results in Fig. 6a-b and Extended Data Fig. 8, where the Pearson correlation coefficient between the ground truth and the predictions from both ReDeconv and MuSiC exceeds 0.99. Therefore, within these synthetic bulk RNA-seq data, the impacts of Type-I, Type-II issues, and their combined effects on deconvolution are clearly discernible.

On the contrary, for the real bulk RNA-seq data, significant disparities exist in the mean expression and standard deviation of the cells in the reference and mixture samples. The outcomes in Figures 4d-e were noticeably influenced by Type-III issues. It's also evident that Type-III issues significantly impact CIBERSORTx, BayesPrism, and MuSiC when the expression profiles of cells in the reference and mixture samples exhibited larger differences (as shown in Fig. 6c, Extended Data Fig. 10b, c, e).

3. In Figure 4a, ReDeconv shows a perfect prediction of the ground truth fractions with minimal deviation. However, in Figure 5b (first column), where deconvolution of the same dataset is performed, a relative error of around 10% is reported. This inconsistency needs to be explained.

Response: We apologize for the confusion. Actually, Fig. 4a and Fig. 5b aligned well with each other. In Fig. 4a, the ground truth percentage for all cell types stood at 16.67%. As a result, a 10% relative error merely indicated an increase or decrease of 1.667% from this ground truth percentage. Upon magnification of the ReDeconv results in Fig. 4a, it became apparent that the findings in Fig. 4a and 5b were consistent. This consistency could be observed in the boxplot and violin plot of ReDeconv results derived from the data in Fig. 4a (Response Fig. 4a, b).

Response Fig. 4 | Detailed examination of the ReDeconv results from Fig. 4a. a. A magnified view of the ReDeconv results from Fig. 4a, demonstrating the consistency between the results of ReDeconv in Fig. 4a and 5b. **b.** Violin plots visualizing the same data in Response Fig. 4a.

4. The authors should benchmark against BLADE deconvolution software (<https://www.nature.com/articles/s41467-021-26328-2>), which also addresses the issue of variance size (Type III).

Response: Thank you for your suggestion to compare with the BLADE algorithm. We have performed a systematic evaluation of BLADE vs. ReDeconv using three synthetic datasets and concluded that BLADE significantly underperformed ReDeconv as summarized in Response Fig. 5. For instance, when BLADE was tested using scRNA-seq data R1 (raw-count) and TPM synthetic bulk RNA-seq data M2, M4, and M22, the outcomes were anticipated to be free of Type-I and Type-II issues. Contrarily, the actual results from BLADE significantly diverged from the ground truth, whereas both MuSiC and ReDeconv yielded highly accurate predictions (Response Fig. 5b and c).

A fundamental difference in model design between BLADE and other methods such as BayesPrism, CIBERSORTx, MuSiC, and ReDeconv lies in the evaluation of the association of expressions of reference and bulk mixtures. BLADE assesses this association on a log-scale, whereas the other four methods examine it in the linear space (not log-scaled). As a result, even

in BLADE's original paper, the Pearson correlation coefficient between the predicted fractions and the ground truth for synthetic bulk averaged at ~ 0.7 (as per Fig. 5b in the BLADE publication). On the other hand, these Pearson correlation coefficients for both MuSiC and ReDeconv consistently approximate 0.99 for the synthetic bulk data we generated.

Overall, while ReDeconv shows potential, the manuscript would benefit from addressing these points to provide a more robust and convincing validation of the method.

Response: Thanks for your supportive remarks and insightful suggestions, which have greatly contributed to enhancing the quality of our manuscript.

Remarks on code availability:

I did not review the code per se, but I did run the website of the software using the test data. The website worked properly, and I got the expected results, within a reasonable frame of time. In principle, it seems a suitable resource for the community.

Response: We sincerely appreciate you taking the time to test our ReDeconv portal and for your positive feedback.

Reviewer #2:

In this manuscript, the authors developed a novel algorithm, ReDeconv, to normalize scRNA-seq data and deconvolute bulk RNA-seq. ReDeconv corrects differentially expressed genes typically misidentified by CP10K method and enhances the accuracy of bulk deconvolution by mitigating gene length effect and modeling the expression variation. The authors performed relatively comprehensive evaluation on bulk RNA deconvolution by comparing with the existing tools and demonstrated its advantages. The algorithm has potential interest and could benefit the community. However, the current version of manuscript has the following drawbacks: 1) it lacks a deeper and comprehensive evaluation of scRNA-seq normalization; 2) It exists several unproved or not well explained key hypothesis; and 3). It is illogical and incoherent in some parts of writing. Below are the detailed comments.

Response: We appreciate your positive comments that our ReDeconv algorithm has potential interests and could benefit the community. We believe we have addressed all your concerns as detailed below.

My major concerns:

1. It seems that CLTS method in ReDeconv is too much simplified in the potential effect from technical noise. The paper 'Droplet barcoding for single-cell transcriptomics applied to embryonic stem cells' demonstrated technical noises were contributed to the scRNA variability. But from Lines 47-48, the authors claimed 'The observed differences in transcriptome size of scRNA-seq data across a broad range of tissues and species suggest that they are inherent to the nature of biology, rather than being artifacts induced by technology-derived effects'. So, this needs more clarification and prove.

Response: We appreciate your comments and agree that CLTS method for scRNA-seq normalization requires further clarification.

As outlined in the paper "*Comparison of transformations for single-cell RNA-seq data*" (*Nat Methods*. 2023 May;20(5):665-672), technical noise is defined as the stochastic effect of capture efficiency, denoted as β . This refers to the probability that an mRNA transcript is captured, and this factor indeed contributes to the variance in the observed transcriptome size across different cells. However, within the framework of CLTS, our focus is on a biological signal: the inherent differences in transcriptome size among various cell types. Assuming the "true transcriptome size" of cell type t is N_t , the expected observed transcriptome size would be βN_t . When performing a linear regression for the average transcriptome size of different cell types across samples, the variance becomes irrelevant as the regression automatically accommodates the possibility of different β values across samples. This argument is based on the assumption that all transcripts within the same droplet are sampled with the same efficiency. While this may not always hold true due to variability in mRNA and primer interactions affecting accessibility, we believe this assumption is satisfactory for our purposes.

On a separate note, CLTS adopts the concept of CP10K/CPM normalization typically used in scRNA-seq data analysis but ensures that the transcriptome sizes of the same cell types remain consistent.

We have added additional clarifications in the revised manuscript (line 47-52).

2a. Follow point 1, Line 112 'cells of the same type demonstrate considerable similarity in (true)

transcriptome sizes.' is not accurate as Extend Data Fig.1 clear show L5 PT CTX have broad range of transcriptome size, which might be resulted from technical noise.

Response: We appreciate your insightful observations and comments. Cells of the same type generally exhibit similar expression profiles, implying that their "true" transcriptome sizes should be comparable. However, not all cells of the same type are always in identical states, such as different phases of the cell cycle, the bursty nature of transcription, etc. leading to potential variance or "biological noise" in their "true" transcriptome sizes. Another potential source of significant biological noise could stem from the incorrect identification of cell types. For instance, L5 PT CTX refers to a specific type of pyramidal neuron located in the fifth layer of the cerebral cortex, but cells classified as L5 PT CTX could potentially encompass other neuron types, such as pyramidal neurons from the sixth layer of the cerebral cortex.

Nonetheless, based on the simple model of capture efficiency that we proposed, if we assume the true number of transcripts is n_i , the variance of observed transcripts would be $\beta(1 - \beta)n_i$. As a result, the coefficient of variation (CV) simply scales with $\frac{1}{\sqrt{n}}$, indicating that the impact of actual technical noise is relatively minor.

As we reasoned in our response to your previous question, we believe that the technical noise was regressed out with a reasonable assumption and the major variance of cell type transcriptome size is likely from the intrinsic biological factors. We hope you would agree.

2b. While one of the purposes for using CP10K is to mitigate the technical noise, how CLTS mitigates technical noise is not clear. It looks like be omitted in the current version by making a strong and unproved hypothesis.

Response: We completely agree with you that CP10K is capable of reducing technical noise related to capture efficiency. However, as outlined in the paper you suggested, "*Droplet Barcoding for Single-Cell Transcriptomics Applied to Embryonic Stem Cells*" (*Cell* 2015 May), it undesirably inflates the CV estimates for each gene by the cell-to-cell variability in total mRNA. Indeed, there is a debate in literature on whether CP10K normalization is good or not (refer to "*Comparison of Transformations for Single-Cell RNA-seq Data*," *Nat Methods*, 2023, and Boeshaghi, et al. 2022, <https://www.biorxiv.org/content/10.1101/2022.05.06.490859v1>).

While CLTS might not reduce technical noise, it does avoid the error introduced by CP10K. Most importantly, a central point of our manuscript is that CLTS rectifies one of the significant drawbacks of CP10K, specifically the Type-1 issues when being used as the reference for bulk RNA-seq deconvolution.

We believe that CLTS does not propose a bold, untested hypothesis, but rather embraces several assumptions we deem logical. A key hypothesis at the heart of CLTS is that the "measured" transcriptome size in cells of the same type should exhibit similarity, a concept fundamentally aligned with CP10K/CPM.

Another hypothesis suggests that for scRNA-seq data derived from cells of the same sample, the raw-counts or "measured" expression values of all genes across all cells should align with their genuine expression values at a **similar ratio**, despite potentially significant levels of technical noise arising from different scRNA-seq methods. This hypothesis should hold true across all reliable scRNA-seq methodologies, as evidenced by the paper "*Characterizing Noise Structure*

in *Single-Cell RNA-seq Distinguishes Genuine from Technical Stochastic Allelic Expression*, *Nat Commun.* 2015; 6: 8687. The study demonstrates that technical noise significantly impacts genes with low to moderate expression. We also demonstrated this point in our response to **Reviewer #1's Question 1a**.

Therefore, even if the raw count of a gene in a single cell may be significantly influenced by technical noise, we can still identify increased gene expression levels at the population level. For example, a gene might display higher expression in L5 cells compared to Astro cells. Similarly, the "measured" transcriptome size means for all cell types in the same sample should maintain a similar ratio to their "true" transcriptome size means. Moreover, for all cell types in different samples, there should be a **strong linear correlation**. CLTS leverages this linear correlation to normalize scRNA-seq data across multiple samples.

3. Based on the description of the method, CLTS is a 'supervised' normalization method since it needs cell type information. In this case, I wondered if CLTS could work on data that cell type is not clear identified? If not, then the cell types identified by conventional CP10K already introduce a bias (claimed by paper), how does CLTS re-correct the potential misidentification of cell type by CP10K?

Response: Thank you for your insightful comments and question. Yes, CLTS does require cell type information, necessitating the application of other methods such as Seurat, Scanpy, scVI, or our own MICA (<https://github.com/jyyulab/MICA>) algorithms to determine the cell type of the scRNA-seq data. CP10K doesn't pose significant challenges for cell clustering. If the marker genes chosen to identify the cell type for each cluster are predominantly expressed in one type of cell and exhibit minimal expression in all other cell types, cell type determination for all clusters under CP10K should not present any issues.

The potential problem with CP10K arises when selecting marker genes that display substantial expression across multiple cell types before normalization. Such genes could be profoundly affected by the CP10K normalization process. Hence, after obtaining cell clusters, we recommend that ReDeconv employs cluster information for normalization. Subsequently, the CLTS normalized data and marker genes should be utilized to identify the cell types of the clusters (refer to the detoured scRNA-seq workflow in Fig. 1a, b and text (line 246-251) in the revised manuscript for further details).

4. Based on the description of the method, CLTS used sample 1 (x1j in Supple Methods) as the base reference. In this case, will the normalized transcriptome size change according to different selections of reference sample? If so, how could CLTS determine which sample is the better representer of 'true' transcriptome size?

Response: We appreciate your insightful question, which was also raised by Reviewer #1. As we detailed in our response to **Question 1b from Reviewer #1**, the relative transcriptome size won't change with different selections of baseline sample and therefore have minimal effects on the CLTS normalization. Further, we have also given solutions in ReDeconv on how to merge similar samples as new possible baseline to improve the cell type coverage and cell numbers for transcriptome size estimations.

5a. Follow point 3, the authors should provide some rationale why they choose these two samples as an example. More samples are needed in order to demonstrate the advantages of CLTS.

Response: The rationale behind choosing these two samples and cell types as examples can be found in the first paragraph of the "**Evaluation of normalization**" in the **Methods section**. In assessing the performance of CLTS, we aim to select samples that meet the following criteria: **(1)** There is significant variation in the transcriptome sizes of the same type of cells across samples, which allows us to observe that the transcriptome sizes of any given cell type in two samples become more similar after CLTS normalization. **(2)** Two cell types should contain a sufficient number of cells (>100), and the transcriptome sizes of cells in these two types should significantly differ, which allows us to show that while CP10K has a scaling issue, CLTS does not. **(3)** The two cell types in the second criterion should also have a significant number of cells in the CosMx data, allowing us to verify if either CLTS or CP10K aligns with the CosMx data. As a result, based on these criteria, we chose these two samples and Astro and L5 IT CTX cells as examples.

In essence, CLTS initially selects a baseline sample B. Then, for any other sample S, CLTS identifies a ratio and uses it to normalize the expression profiles of cells in sample S. This process ensures that the mean transcriptome sizes of each cell type in sample S align with those of the same cell types in the baseline sample B. The key advantage of CLTS is its capacity to minimize the issue of sequencing depth while preserving the distinct transcriptome sizes of various cell types. Consequently, using just two samples as examples should be adequate to illustrate the benefits of CLTS. Nonetheless, we also expect that the differential expression genes (DEGs) between various cell types should not exhibit significant variation in CLTS normalized scRNA-seq data with more samples.

In response to your suggestions, we applied CLTS to normalize scRNA-seq data across four samples. The results indicated that CLTS effectively equalized the mean transcriptome size of the same cell type, resulting in more similar gene expressions of the same type of cells across different samples. A comparison of gene expression in L5 cells and Astro cells revealed a high overlap of DEGs (up-regulated in L5 cells) obtained from CLTS normalized four-sample and two-sample scRNA-seq data, respectively.

We have added the new results (line 240-245 and Supplementary Fig. 5) in our revised manuscript.

5b. And Extend Data Fig. 1b, c needs more explanations and clearer annotation.

Response: We computed the Pearson correlation coefficient of the mean cell type transcriptome size for any two samples that share a minimum of four cell types, each with at least 10 cells in both samples. Extended Data Fig. 1b visually represents the Pearson correlation r for all pairs of lung cancer samples, whereas Extended Data Fig. 1c depicts the Pearson correlation r for all pairs of mouse brain samples.

We have expanded the figure legends and revised the figure annotation to provide more clarity.

6. Line 65, the author mentioned SCnorm and SCTransform also susceptible to the same scaling effect like CP10k. It is better to make a comprehensive comparison by including them in main figs or supple figs.

Response: We thank you for the constructive suggestion and have performed the comparison of CLTS with scNorm and scTransform.

When assessing the relative transcriptome sizes of various cell types (relative to the transcriptome size of Astro cells) in both raw-count and different normalization methods, it becomes clear that CLTS-normalization does not alter the relative transcriptome sizes across all cell types for both samples. In contrast, scNorm (assuming two samples are in identical and varying conditions) and

scTransform significantly change the relative transcriptome sizes of different cell types, indicating a scaling effect. This scaling effect of scNorm and scTransform is also noticeable on individual genes.

The scNorm is notably time-consuming. When tested with the scRNA-seq data set O1_2, (refer to new Fig. 2d, Extended Data Fig. 2a-c), which comprises approximately 50k cells, scNorm was unable to complete within a seven-day period. As a result, a random subset of about 5k cells from the O1_2 data set was chosen for a comparative analysis of different normalization methods. The execution time of scNorm in this scenario was approximately 4.5 hours. Subsequent figures highlight two key observations: **First**, the normalizations from CLTS, CP10K, scTransform, and scNorm_1cd (which considers two samples in identical conditions) can align gene expressions in two samples to a similar distribution. However, the scNorm_2cd normalization (which considers two samples in different conditions) cannot. **Second**, besides CLTS, all other normalization methods exhibit issues related to the scaling effect.

We have added results of comparing different scRNA-seq data normalization methods (line 360-363 and new Supplementary Fig. 6) in our revised manuscript.

New Supplementary Fig. 6 | Analysis of how scaling effects influence scRNA-seq data normalization. a-b, present the mean cell type transcriptome sizes relative to Astrocyte cells in Sample-I and Sample-II under raw-count, CLTS, scNorm (assuming two samples are in identical and varying conditions), and scTransform normalized scRNA-seq data. c-f, exhibit the expression levels of genes Plcb1(c), Ntm (d), Cpe (e), and Mt-Atp6 (f) in the L5 and Astrocyte cells from of mouse brain Sample_I and Sample_II. Data is shown for raw-count, CLTS, CP10K, scNorm, and scTransform normalized

7. Lines 98-100, ‘Consistent with previous reports, we observed that cells of a single type in the same sample exhibit roughly the same transcriptome size, while the transcriptome size varies significantly across different cell types’. It is not clear the meaning of ‘a single type in the same sample exhibit roughly the same transcriptome size’.

Response: We apologize for the typos and have fixed as below: “*within a single sample, the transcriptome size will vary significantly across different cell types, but cells of the same type will have transcriptomes that are similar in size*” (line 138-140).

8a. It looks like Fig. 1b missing many types. Could authors further clarify this?

Response: As we could only show common cell types (cell type names are exactly the same) in both mouse and human scRNA-seq data, we have to miss some cell types in Fig. 1a because the difference between human and mouse scRNA-seq data.

8b. The definition of Type I, II, III issues is not clearly defined in the current version.

Response: Due to space limitations, we relocated extensive content related to Type I, II, and III issues to the **Methods** section in the **Supplementary Information**, as detailed below.

In a mixture sample with bulk RNA-seq data, the RNAs of each gene originate from various cell types. Thus, for any given gene, such as gene i , if we determine the gene's expression in the mixture sample and the expression means ($\mu_{i1}, \mu_{i2}, \dots, \mu_{in}$) of the gene across all cell types, we should be able to establish equations (1). This concept forms the fundamental basis for using references to ascertain the fraction of different cell types present in the mixture samples.

$$\begin{aligned}
 f_1 u_{11} + f_2 u_{12} + \dots + f_n u_{1n} &\approx x_1 \\
 f_1 u_{21} + f_2 u_{22} + \dots + f_n u_{2n} &\approx x_2 \\
 &\dots \dots \dots \\
 f_1 u_{m1} + f_2 u_{m2} + \dots + f_n u_{mn} &\approx x_m
 \end{aligned}
 \tag{1}$$

Type-I issues arise from the application of inappropriate normalization methods, such as CP10K and CPM, to scRNA-seq data used as a reference for deconvolution. This primarily addresses the sequence depth problem. For instance, because CP10K would amplify the expression of genes in cells with smaller transcriptome sizes, it could alter the expression means of genes in cell type 1 from μ_{i1} to $2\mu_{i1}$. To maintain the balance of equations (1), the fraction of cell type 1 would be adjusted from f_1 to $f_1/2$. Therefore, the influence of Type-I issues on deconvolution can be elucidated by equations (2). As illustrated in Fig. 5a, Extended Data Fig. 6, there is a clear agreement between the results and our mathematical analysis.

$$\begin{aligned}
 (f_1/r_1)(r_1 u_{11}) + (f_2/r_2)(r_2 u_{12}) + \dots + (f_n/r_n)(r_n u_{1n}) &\approx x_1 \\
 (f_1/r_1)(r_1 u_{21}) + (f_2/r_2)(r_2 u_{22}) + \dots + (f_n/r_n)(r_n u_{2n}) &\approx x_2 \\
 &\dots \dots \dots \\
 (f_1/r_1)(r_1 u_{m1}) + (f_2/r_2)(r_2 u_{m2}) + \dots + (f_n/r_n)(r_n u_{mn}) &\approx x_m
 \end{aligned}
 \tag{2}$$

Type-II issues stem from the application of mismatched normalization to the scRNA-seq and bulk RNA-seq data used for deconvolution. This primarily pertains to the gene length normalization. As demonstrated in Supplementary Fig. 1, the raw-count expression of genes in the scRNA-seq data is not linked to gene length, while in the total bulk RNA-seq data, it is. Consequently, if we employ raw-count scRNA-seq and bulk RNA-seq data (or TPM scRNA-seq and bulk RNA-seq

data) for deconvolution, our solution will be dictated by equations (3), rather than being derived from equations (1). This elucidates how Type-II issues influence deconvolution. (L_i represents the gene length of gene i)

$$\begin{aligned}
 f'_1 u_{11} + f'_2 u_{12} + \dots + f'_n u_{1n} &\approx L_1 x_1 \\
 f'_1 u_{21} + f'_2 u_{22} + \dots + f'_n u_{2n} &\approx L_2 x_2 \\
 &\dots \dots \dots \\
 f'_1 u_{m1} + f'_2 u_{m2} + \dots + f'_n u_{mn} &\approx L_m x_m
 \end{aligned} \tag{3}$$

Type-III issues pertain to the robustness of deconvolution models. The expression value of each gene within a cell of a given type is not constant. Thus, the expression of a gene in the same cell type in the reference (scRNA-seq data) and mixture samples (bulk RNA-seq data) should exhibit some differences, which can impact the accuracy of fraction predictions. To address this issue, we **first** select signature genes with more stable expressions. **Secondly**, we incorporate the expression variance of these signature genes into our new computational model designed to determine cell type fractions in the mixture samples (please refer to equation (5)).

$$\begin{aligned}
 f(f_1, f_2, \dots, f_n) &= \prod_{i=1}^m \left(\frac{1}{\sqrt{2\pi c^2 \sigma_i^2}} \exp\left(-\frac{(x_i - c\mu_i)^2}{2c^2 \sigma_i^2}\right) \right) \\
 &= \prod_{i=1}^m \left(\frac{1}{\sqrt{2\pi c \sum_{t=1}^n f_t \sigma_{it}^2}} \exp\left(-\frac{(x_i - \sum_{t=1}^n f_t \mu_{it})^2}{2c \sum_{t=1}^n f_t \sigma_{it}^2}\right) \right)
 \end{aligned} \tag{5}$$

8c. Are these biologically related issues or technically related issues?

Response: Type-I and Type-II issues stem from incorrect normalization of scRNA-seq and/or bulk RNA-seq data. Utilizing CLTS for scRNA-seq data normalization and TPM for bulk RNA-seq data normalization can effectively mitigate these issues. Type-III issues are a result of both biological and technical noise.

8d. For example, how type I issue reflects false-positive in deconvolution scenario?

Response: The goal of deconvolution is to determine the proportions of different cell types in mixed samples, which is not a classification problem. Defining a prediction as true positive or negative is not straightforward, particularly in terms of determining the threshold difference between the predicted fraction and the ground truth fraction that would constitute a true positive. Consequently, we didn't evaluate how Type-I issues affect false-positive outcomes in deconvolution scenarios.

9. Line 221, Supplementary Fig.2c. Could author provide a real world example to prove?

Response: We sincerely appreciate your suggestions. As a response, we have referenced a paper titled "*Transcript length bias in RNA-seq data confounds systems biology. Biol Direct 4, 14 (2009)*", which elucidated the need for gene length normalization in the analysis of raw-count bulk RNA-seq data (reference 22). Additionally, we have cited the paper "*Quantitative single-cell transcriptomics. Brief Funct Genomics 17, 220-232 (2018)*" that provided an in-depth explanation of the generation of scRNA-seq data (reference 23). These references support our description.

For a comprehensive understanding of scRNA-seq or bulk RNA-seq data under a specific protocol, it is imperative to examine the respective protocol pertaining to sample preparation and RNA sequencing.

10. Line 227, 'the expression of each gene in each cell of any given cell type is a random event' needs to be referred or prove.

Response: We highly appreciate your concerns about the description of type-III issues. We concur that the text used to describe the gene expression in each cell in the initial manuscript was not precise and could have been misleading. In response, we have refined the text in new line 118-119. The main point we aimed to underscore is that for both scRNA and bulk RNA data, the expression value of each gene adheres to certain distributions (refer to "*The shape of gene expression distributions matter: how incorporating distribution shape improves the interpretation of cancer transcriptomic data*". *BMC Bioinformatics* 21, 562, 2020) across a group of cells or samples. Existing methods often depend solely on the mean to characterize the entire distribution, thereby not fully capitalizing on the additional information offered by large datasets. Conversely, ReDeconv incorporates both the mean and variance into its model. This enhances the precision of our gene distribution descriptions and results in improved deconvolution outcomes.

11. It would be better to move the workflow of the algorithm at the beginning.

Response: Following your constructive suggestion, we have moved the workflow to Fig. 1 in the revised manuscript.

My minor comments:

1. Line 45, references are needed for multiple scRNA-seq datasets.

Response: We deeply appreciate your comment. In response, we have incorporated the following three references that support our observations (reference 2-4).

1. Islam, S. et al. Characterization of the single-cell transcriptional landscape by highly multiplex RNA-seq. *Genome Res* 21, 1160-1167 (2011).
2. Jonasson, E. et al. Total mRNA Quantification in Single Cells: Sarcoma Cell Heterogeneity. *Cells* 9 (2020).
3. Yao, Z. et al. A taxonomy of transcriptomic cell types across the isocortex and hippocampal formation. *Cell* 184, 3222-3241 e3226 (2021).

2. More detailed descriptions for Supple Fig. 1 are needed. Current explanation is hard to follow

Response: Supplementary Fig. 1a (new Supplementary Fig. 2a) shows the scatter plots of mean cell type transcriptome sizes between lung cancer sample EBUS_49 and other lung cancer samples in GSE131907. Supple Fig. 1b (new Supplementary Fig. 2b) shows the scatter plots of mean cell type transcriptome sizes between mouse sample 352356 and other mouse samples in the scRNA-seq dataset from the Allen Brain Map.

We have updated the figure legend to make it clear.

3. It would be better to only have one terminology system for supplementary materials (extend data Fig in line 106, extended Fig , line 98; supplementary fig Line 107).

Response: Apologies for any confusion caused by the classification of figures into main, extended data, and supplementary categories. We followed the formatting of Nature Biotechnology (<https://www.nature.com/nbt/submission-guidelines/aip-and-formatting#extended-data-figures>) and other *Nature* journals, which allows maximum of 10 Extended data figures. We had to put less important figures as supplementary figures.

To avoid confusion, we have put Supplementary Figures into “Supplementary Documentation” file together with supplementary information and supplementary tables.

4. Same r value in both Fig. 1a and 1c. Is it a typo?

Response: No, this is not a typo. As our new method for normalization is mainly amplifying the expression profile of each cell in any given sample with a fixed ratio, it will not alter the linear correlation of mean of cell type transcriptome size in any two samples.

5. More clear explanations are needed for $u_{icg}/a_i + (v_{max} - b_j/a_i)/G$.

Response: As for any cell type j , its transcriptome size mean in sample i , x_{ij} , and in baseline sample 1, x_{1j} , have the linear relation: $x_{ij} = a_i x_{1j} + b_i$, $1 \leq j \leq n$. Applying the equation $x_{ij}/a_i - b_i/a_i$ will equalize the mean transcriptome size of cell type j in sample i to that in the baseline sample 1. In order to normalize each gene of each cell in sample i , we use the formula $\frac{u_{icg}}{a_i} + \frac{-b_i/a_i}{G}$, where G represents the number of genes in the scRNA-seq data. Essentially, the expression profile of each cell in sample i will be adjusted by a ratio of $\frac{1}{a_i}$. The shift $-b_i/a_i$ will then be evenly distributed to all genes. By adding $\frac{v_{max}}{G}$ to the new expression value of each gene of each cell in sample i , we can avoid the negative expression values for the CLTS-normalized data.

We have added more explanations in the **Methods Section** (the second paragraph).

6. Fig. 2h needs more explanations.

Response: We highly appreciate your comment. In brief, the CosMx is a cutting-edge image-based spatial transcriptomics platform. It can detect the expression of 1,000-5,000 genes at single-cell resolution across multiple regions (FOV) in a single sample section. Nanostring has publicly available CosMx datasets for both human and mouse brains, which we used to validate the accuracy of DEG identified by our CLTS approach. We selected *Plcb1*, *Tnik*, and *Rora* as examples and examined their expression values in L5 and AS cells using the CosMx data. As shown in Fig. 2h (new Fig. 3h), the white circles in three FOVs (on the left) represent AS cells, and the blue dots represent the expression of *Plcb1*, *Tnik*, and *Rora* (from top to bottom). Similarly, the white circles in the same three FOVs (on the right) represent L5 cells, and the red dots represent the expression of *Plcb1*, *Tnik*, and *Rora* (from top to bottom). It is clear that all three genes exhibited higher expression in L5 cells compared to AS cells, which is consistent with CLTS approach.

We have revised the legend of Fig. 2h (new Fig. 3h) and added detailed explanations about CosMx data in the figure legend.

Reviewer #3:

Reviewer #4:

Response: We thank both Reviewers #3 and #4 for their co-reviewing our manuscript.

Point-by-point response to reviewers' comments

Reviewer #1:

We appreciate the authors' efforts to implement the suggested improvements and clarify the manuscript. The changes have significantly benefited the overall quality; however, we believe some points still require further clarification:

Response: We greatly appreciate all your comments, which have significantly contributed to the improvement of our manuscript.

Regarding my first comment (1a in author's response) about comparing different protocols, we do see the authors' point that the New Fig. 2b, shows a correlation between protocols, although they are versions of the same protocol. The correlation of the different protocols shown in the New Supp. Fig 3, partly address the question about other protocols, but we think to correctly do it the authors should show a plot similar to New Fig. 2b with the 9 different cell types sequenced with the different protocols (with the exception of the inDrops one), for both PBMC1 and 2.

Response: Based on your suggestion, we have updated Supp. Fig. 3 (excluding inDrops in both samples). We only showed cell types with at least 10 cells in the data sets of both protocols. The scatter plot provides detailed insights into the strong linear correlations between the average transcriptome sizes of different cell types across various protocols.

Regarding comment 1b, which discusses the choice of reference sample, we agree with the authors that the correlation remains consistent regardless of the selected reference. However, we disagree in that this proves that it has no impact at all. For instance, in Response Fig 1c and d (Sample III and Sample IV as reference respectively), in the left plots (comparison of Sample I and II), the pink sample (I think corresponding to Set, although it's difficult to differentiate between all pink samples in the legend) is the third with the highest transcriptome size in d (see red arrow in the attached image), while in c the same sample is the 7th (or more) sample in transcriptome size (blue arrow in the attached document). If the statement of sample size not being affected by technical artefacts, and only by cell type holds true, this couldn't be the case, so either that statement needs to be revised or this strategy is affecting the processing of the data. Authors need to clarify this matter, as their normalization method is based on that statement.

Response: We sincerely apologize that we made a mistake in providing the wrong legend for Response Fig.1 in our previous submission, which might have caused a misinterpretation. In the previous Response Fig. 1, we discussed whether the baseline selection has an impact on the CLTS normalization. In the figure, we intended to demonstrate that for a set of samples exhibiting a strong linear correlation between the cell type transcriptome size means of any two samples, employing any sample from the set as a baseline for CLTS-normalization would not affect or only slightly impact the relative cell type transcriptome size means of any sample within the set. For instance, the left-most figure of the previous Response Fig. 1 illustrated the Pearson correlation of cell type mean transcriptome size in Sample_I normalized using Sample_I (y-axis) as the baseline and that normalized using Sample_II (x-axis) as the baseline. Similarly, the central figures present the results using Sample_I as the baseline versus using Sample_III as the baseline. The right-most figure displays the results using Sample_I as the baseline versus using Sample_IV as the baseline. Previous Response Fig. 1b, c, and d were for Sample_II, III, and IV respectively. We have updated **Response Fig. 1** below with the correct figure legend.

Following your thought, we have compared the mean transcriptome size of cell types in Sample I vs. Sample II after normalization using Sample III or Sample IV as the baseline, and the results indicated that the order of cell types in the two cases remains unchanged (**Response Fig. 2**).

We agree with you that selecting different samples as baselines for CLTS normalization can influence the results, although this impact is generally minimal. Essentially, during CLTS normalization, for any given sample, once a baseline is selected, the transcriptome size mean of any cell type i , T_i , will be normalized to $c_1 T_i + c_2$, where c_1 and c_2 are determined by the fixed sample and the baseline. For any two distinct cell types i and j that do not have the same transcriptome size mean, $\frac{T_i}{T_j} = \frac{c_1 T_i + c_2}{c_1 T_j + c_2}$ if and only if $c_2 = 0$. Consequently, the chosen baseline may yield different values for c_2 , implying that CLTS normalization could influence the relative transcriptome size means of cell types in any sample. Given that c_2 is typically small when the linear correlation of cell type transcriptome size means between the fixed sample and the baseline is strong, this impact is relatively minor in our application (Response Fig. 1b). As all cell type transcriptome size means within the same sample undergo the same linear transformation, it's clear that CLTS normalization won't change the order of these means in any given sample.

Response Figure 1 | (Previous Response Fig. 1 with correct legend) Comparison of mean cell type transcriptome sizes in mouse sample using different normalization baselines. a-d, Mean cell type transcriptome sizes in Mouse Sample-I (a), Sample-II (b), Sample-III (c), and Sample-IV (d), respectively, utilizing varied baselines for CLTS normalization; y-axis represents cell type transcriptome size means in the data using Sample-I as baseline while x-axis represents those in the data using Sample-II, III, or IV as baseline.

As for the other points we raised, we appreciate the authors' efforts to address them, and their responses are satisfactory.

Response: Thank you very much.

Reviewer #2:

The author addressed most of my concerns and clarified some ambiguities. However, a few points are still needed to be clarify.

1. Point #3 remains unclear. I am still uncertain about how so many genes are misidentified by CP10K, yet this does not impact cell type annotation. Could the authors provide a profile of the marker genes identified using CP10K and corrected by CTLS? How many of these marker genes are affected? If most marker genes are impacted, it raises questions about the reliability of cell type annotation for CTLS correction. Conversely, if only a few are affected, how does CTLS enhance downstream analyses such as trajectory analysis, GO, GSEA, annotation, cell-cell interaction analysis, and metacell analysis, as shown in Figure 1a?

Response: We appreciate your insightful points and questions. In our study, we used astrocytes and L5 IT cells as examples to demonstrate the significant impact of CP10K normalization on identifying differentially expressed genes (DEGs) between these two cell types. For cell type annotation, researchers often rely on established cell type marker genes, which are typically highly expressed in a specific cell type and are generally unaffected by CP10K normalization. However, certain marker genes, which may be highly expressed in more than one cell type, can have their relative expression patterns influenced by CP10K.

For instance, we performed CP10K-normalized clustering using Seurat on the dataset "O7_matrix_446701_410107_raw_count_smallDataSet.tsv" and identified six clusters (**Response Fig. 3a**). These clusters corresponded well with the cell type annotations (**Response Fig. 3b**) from the original study (Trygve E Bakken et al., *Nature* 2021 Oct;598(7879):111-119. PMID: 34616062), with cluster assignments as follows: 0 – L5 IT, 1 – L2-3 IT, 2 – L6b, 3 – Oligo, 4 – Lamp5, and 5 – Astro. The original study (Supplementary Table 5, *Nature* 2021, PMID: 34616062) used several known marker genes for these annotations (**Response Fig. 3c**).

We then evaluated the expression patterns of these marker genes under both CP10K and CLTS normalization (**Response Fig. 4**). As expected, most marker genes were predominantly expressed in a single cell type and were unaffected by CP10K. However, the expression pattern of *St6galnac5*, a marker gene for L5 IT cells, was notably altered by CP10K. In CLTS-normalized data, *St6galnac5* showed the highest expression in L5 IT cells (\log_2FC of L5 IT vs. Astro = 1.68), its correct cell type, whereas in CP10K-normalized data, its highest expression appeared incorrectly in Astro cells (\log_2FC of L5 IT vs. Astro = -0.67). Thus, CLTS normalization corrected the expression pattern of *St6galnac5* for L5 IT cells. Despite CP10K's effect on a few marker genes, we were still able to accurately annotate cell types across clusters using known marker

genes. Overall, CTLS is unlikely to affect cell type annotation but can adjust the relative expression patterns of some marker genes, especially those expressed across multiple cell types.

While CP10K has minimal effects on cell type annotation, it can significantly impact relative gene expression patterns or DEG identification across cell types, which in turn affects downstream analyses like GO and GSEA functional enrichment. This impact, as we demonstrated in the astrocyte vs. L5 IT cell comparison (**Fig. 3 and ED Fig. 3** in the revised manuscript), is particularly pronounced when there are substantial differences in transcriptome sizes among cell types.

2. Figure 2 demonstrates that while transcriptome sizes vary across different cell types, their relative positions along the linear line are similar. If we first quantile the cell sizes and then normalize based on these quantiles, could this approach reduce the dependence of CTLS on cell type annotation. Then CTLS could perform as an alternative to CP10K and mitigate issues that arise from cell type annotation errors associated with CP10K?

Response: We are incredibly grateful for your constructive suggestions on reducing CTLS's reliance on CP10K-based clustering and cell type annotation. In fact, we have been actively exploring this approach and have tried several strategies, though we did not include these in the manuscript.

We appreciate your suggestion regarding the partitioning of cells by quantile transcriptome sizes, which is indeed an intriguing approach. However, we anticipate that this method could face two potential challenges. First, because transcriptome sizes of various cell types can overlap, each quantile may contain a mixture of cell types. This could be problematic if matched quantiles in two samples contain different proportions of cell types, as frequently observed in datasets such as the mouse brain dataset used in our study (**Response Fig. 5**). Second, if a particular cell type is absent in one sample, matching quantiles may result in the normalization forcing dissimilar cell types in each sample to have similar mean transcriptome sizes. For example, if we assume two samples containing ten cell types in total, with T1 and T2 as the cell types with the largest mean transcriptome sizes (15k and 12k, respectively), sample-1 might lack T1 while sample-2 includes all ten types. In such a case, normalization using matched quantiles could inadvertently equate cell type T1 in sample-2 with cell type T2 in sample-1.

To address these issues, we propose an alternative approach: performing cell clustering on raw count scRNA-seq data, followed by CTLS normalization. Current clustering algorithms, such as Seurat, use Euclidean distance to assess the similarity of cell expression profiles, requiring CP10K normalization to counteract sequencing depth effects. To overcome this limitation, we have been developing scMINER, a mutual information-based framework for cell clustering, network analysis, and hidden driver inference (Ding, L. et al. scMINER: a mutual information-based framework for identifying hidden drivers from single-cell omics data. *Res Sq* 2023, PMID: 36747870). Within scMINER, we created MICA (<https://github.com/jyyulab/MICA>), which leverages mutual information to calculate expression similarity among cells as a distance metric for clustering. Using this approach, scMINER-MICA can bypass CP10K normalization and perform clustering directly on raw count data. For instance, we applied MICA to the raw-count O7 scRNA-seq dataset, and the resulting clusters aligned perfectly with those derived from CP10K-normalized data in Seurat (**Response Fig. 6**). This approach effectively eliminates CP10K dependence and addresses additional CP10K-related issues.

We did not include this approach in our initial submission, as the scMINER manuscript is currently under revision, though it is available on bioRxiv. In the revised manuscript, we have added a paragraph (lines 429–439) in the Discussion section to outline this potential CP10K-independent approach, alongside our recommendations for integrating ReDeconv with Seurat, Scanpy, and scMINER for comprehensive scRNA-seq data analyses.

3. Github page is not available.

Response: Thank you for your kind reminder. We have adjusted the visibility settings and made ReDeconv's GitHub page accessible to the public.

4. The CLI and API documentation should be provided more in details. Additionally, please provide a comprehensive tutorial on integrating ReDeconv with Seurat and Scanpy which would facilitate the usage of the tool in the community. (Remarks on code availability)

Response: We sincerely appreciate your constructive suggestions to enhance our ReDeconv software. In the "Documentation" section of the ReDeconv website, we have expanded the four main tabs—"Introduction," "Tutorial – Desktop Program," "Tutorial – Web Portal," and "Demo Scripts"—to include detailed information and tutorials covering the various ReDeconv programs and applications.

Following your suggestion, we have also added examples demonstrating how to use the ReDeconv-GUI and integrate ReDeconv with Seurat, Scanpy, and scMINER-MICA. These examples are included in the 'Examples.zip' file, available in the "Software" section of the ReDeconv website, and referenced under "Code availability" (lines 522-524) in the manuscript.

Additionally, in the revised manuscript, we have added a paragraph (lines 429-439) in the Discussion section recommending the integration of ReDeconv with Seurat, Scanpy, and scMINER for scRNA-seq data analysis.

Reviewer #3:

I co-reviewed this manuscript with one of the reviewers who provided the listed reports. This is part of the Nature Communications initiative to facilitate training in peer review and to provide appropriate recognition for Early Career Researchers who co-review manuscripts. (Remarks on code availability)

Reviewer #4:

Response: We thank both Reviewers #3 and #4 for their co-reviewing our manuscript.

CMouse Sample III: Pearson Correlation $r=1.00$ **d**Mouse Sample IV: Pearson Correlation $r=1.00$